# CONTINUAL FINE-TUNING WITH PROVABLY ACCURATE AND PARAMETER-FREE TASK RETRIEVAL

## ABSTRACT

Continual fine-tuning aims to adapt a pre-trained backbone to new tasks sequentially while preserving performance on earlier tasks whose data are no longer available. Existing approaches fall into two categories which include input- and parameter-adaptation. Input-adaptation methods rely on retrieving the most relevant prompts at test time, but require continuously learning a retrieval function that is prone to forgetting. Parameter-adaptation methods instead use a fixed input embedding function to enable retrieval-free prediction and avoid forgetting, but sacrifice representation adaptability. To combine their best strengths, we propose a new parameter-adaptation method that enables adaptive use of input embeddings during test time with parameter-free retrieval. We derive task-retrieval error bounds for a clustering-based, parameter-free paradigm, providing theoretical guarantees that link low retrieval error to structural properties of task-specific representation clusters, revealing a fresh insight into how well-organized clustering structure will enable reliable retrieval. Motivated by this insight, our method is designed with two key components: (i) an adaptive module composition strategy that learns informative task-specific updates to preserve and complement prior knowledge, and (ii) a clustering-based retrieval mechanism that captures distinct representation signatures for each task, enabling adaptive representation use at test time. Extensive experiments show that these components work synergistically to improve retrieval and predictive performance under large shifts in task semantics.

## 1 INTRODUCTION

Continual learning (CL) is a classical challenge in machine learning (ML) where a model must learn from a sequence of tasks that arrive one at a time and with their own distinct data distributions. A key constraint in this setting is that data from previous tasks cannot be retained once new tasks arrive, obscuring the overall view of all task distributions. This leads to catastrophic forgetting (van de Ven et al., 2022; McCloskey & Cohen, 1989) in which learning new tasks interferes with previous knowledge. Classical approaches often mitigate forgetting via regularization (Titsias et al., 2019; Pan et al., 2020), replay buffers (Buzzega et al., 2020; Cha et al., 2021; Chaudhry et al., 2019), or pruning (Hung et al., 2019; Golkar et al., 2019). However, these strategies either update/prune large shared models, or maintain/retrieve from sizable parameter sets or representative samples for each task, which struggle to scale when task sequences become long with increasing shifts in semantics.

Against these limitations, the recent rise of large foundation models (Devlin et al., 2019) has opened up new possibilities for CL designs, enabling a paradigm where a pre-trained backbone is frozen to ensure knowledge retention. During training, task-specific updates are incorporated via either input-adaptation or parameter-adaptation methods. To elaborate, input adaptation refers to adaptation methods that append learnable parameters (i.e., prompts) to the tokenized input sequences (e.g., sequences of image patches). Examples include prompt-tuning methods (Lester et al., 2021). Parameter adaptation refers to methods that add low-complexity trainable parameter blocks to its pre-trained parameters. Examples include LoRA-based methods (Hu et al., 2021). Learning these units requires significantly smaller memory footprint and computation cost than maintaining and updating large sets of shared parameters with sophisticated regularization.

For example, RanPAC addresses the issue of forgetting via (1) fine-tuning a low-rank adaptation (LoRA) unit using data from the first task; and (2) using it to compute input and class embeddings

across all tasks (McDonnell et al., 2023). A parameter-free classification model (e.g., nearest neighbor or linear discriminant analysis) is then employed to map unseen inputs to their most probable classes. Its use of a pre-tuned set of adaptation parameters (using data from the first task) for inputs across all tasks however limits its adaptability in real-world applications involving significant semantic gaps across tasks (Kim et al., 2024) (see Section 5.2).

Alternatively, retrieval-based continual fine-tuning frameworks often adopt a two-stage strategy: (1) learning task-specific fine-tuning parameters to mitigate interference between tasks (Wang et al., 2022a;b;c; 2023a; McDonnell et al., 2023); and (2) retrieving the appropriate parameter set at inference time to handle previously unseen inputs. For instances, L2P maintains a shared pool of prompts to compartmentalize task-specific knowledge and optimizes a prompt-selection mechanism to retrieve relevant prompts for each input (Wang et al., 2022c). This leads to several generalizations such as separating global and local prompts (Wang et al., 2022b), using prompt ensemble to enhance knowledge transfer (Wang et al., 2023a), or maintaining a growing prompt pool (Smith et al., 2023; Wang et al., 2023a; 2022a). While their learnable retrieval mechanisms help mitigate knowledge interference and enable flexible test-time adaptation, these methods retain the risk of forgetting because the parameters of their retrieval modules must be continuously updated as new tasks arrive.

**Contribution.** To alleviate key issues with the lack of representation adaptability in parameter-free classification (McDonnell et al., 2023) and forgetting in the retrieval-based methods, we develop PROTEUS which is a provably accurate, parameter-free task retrieval framework for continual fine-tuning systems. It achieves provably low retrieval error rate and hence superior performance compared to existing state-of-the-arts. This is enabled by the following contributions:

**1. Provably Accurate and Parameter-Free Retrieval.** We develop a theoretical analysis of retrieval error rates for a broad class of parameter-free, signature-based representation retrieval methods. Our framework characterizes retrieval error in terms of the clustering structure of the distributions of signature patterns (see Definition 3.1) created for different task-specific adaptation modules. This provides a principled foundation applicable across diverse retrieval-based CL methods. Concretely, we study algorithms that learn distinct signature patterns from each task's input embeddings to capture both intra- and inter-task variability. These signature patterns serve as keys to retrieve the most relevant representation adaptation modules for each input during test time. The retrieval method is parameter-free, and its error rate is provably controlled by statistical properties of the signature clusters and is shown to be vanishingly small (Section 3.2).

**2. Adaptive Knowledge Transfer and Discovery.** Motivated by the theoretical insight that retrieval error rates are controlled by the clustering properties of signature patterns, we develop an adaptive knowledge transfer and discovery method designed to improve cluster separation (see Definition 3.3) and thus reduce retrieval error. This is achieved via learning new representation components that complement those from previous tasks. This in turn helps segregate new representation attributes from old ones and creates more distinct clusters of representation signatures for new tasks while preserving knowledge transferability. As a result, each task-specific adaptation module and its induced input embedding or representation distributions become easier to discriminate and retrieve during test-time inference, ensuring low retrieval error and improving predictive performance (Section 4).

**3. Empirical Studies.** We demonstrate empirically that the integrated synergy between our adaptive knowledge transfer and parameter-free retrieval mechanism with provably low error rate significantly enhances the performance and scalability of continual fine-tuning. As reported in our empirial studies, our proposed method PROTEUS establishes new SOTA results on a wide variety of prominent class-incremental CL benchmarks, including the highly challenging Visual Task Adaptation Benchmark (VTAB) featuring diverse task sequences large semantic gap (Zhai et al., 2019). It achieves up to 57% and 30% gains in retrieval and classification performance while attaining the best top-1 average forgetting metric, which corroborates our theoretical insights (Section 5).

## 2 PROBLEM FORMULATION AND EXISTING LITERATURE

**Problem Formulation.** Continual fine-tuning (CFT) aims to adapt a pre-trained model to a sequence of tasks whose training datasets $D_1, D_2, \ldots, D_m$ arrive sequentially and cannot be retained due to practical constraints. The key challenge is how to design resilient adaptation methods that adapt the CFT system to new task data effectively while preventing it from forgetting past solution expertises. This can be addressed via a retrieval-based approach: (1) learning task-specific adaptation that

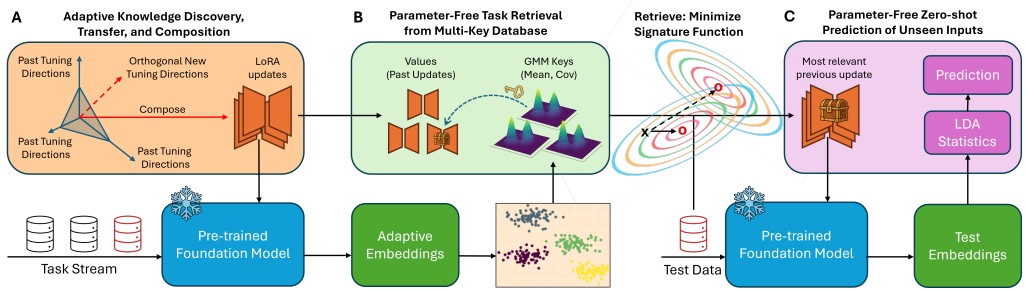

Figure 1: The overall workflow of PROTEUS. **A:** A LoRA unit is optimized for each task as a combination of previous tuning directions and new orthogonal components to capture task-specific information (Section 4). **B:** A database of values (past update directions) and multi-keys (GMM parameters fit to each task embedding distribution) is updated. At test time, this database is used to retrieve most informative past updates per test input (Section 4). **C:** LDA prediction (Section 4).

encodes solution for each task while minimizing interference with prior tasks; and (2) learning a retrieval mechanism to select and compose the most relevant expertise for each test input.

Let $\Delta\boldsymbol{\omega}_k$ denote the set of adaptation parameters or expertises learned for the $k$-th task's dataset $D_k$, and $\boldsymbol{M}_{k-1} = (\Delta\boldsymbol{\omega}_1, \Delta\boldsymbol{\omega}_2, \ldots, \Delta\boldsymbol{\omega}_{k-1})$ represent the database of learned expertises with respect to the first $k-1$ datasets. Upon the arrival of task $k$ with dataset $D_k$, we learn a new set of adaptation parameters $\Delta\boldsymbol{\omega}_k$ and update the parameter $\boldsymbol{\theta}$ of a retrieval score function $\gamma_{\boldsymbol{\theta}}$ via minimizing:

$$\Delta\boldsymbol{\omega}_k, \boldsymbol{\theta} = \underset{\Delta\boldsymbol{\omega}'_k, \boldsymbol{\theta}'}{\arg\min} \left( L\left(D_k, \boldsymbol{M}'_k\right) + \lambda_1\gamma_{\boldsymbol{\theta}'}\left(D_k, \boldsymbol{M}'_k\right) + \lambda_2 R\left(\boldsymbol{M}'_k\right) \right) \quad \text{where}$$

$$\boldsymbol{M}'_k = \text{retrieve}_{\boldsymbol{\theta}}\left(D_k, \boldsymbol{M}_{k-1}\right) \cup \Delta\boldsymbol{\omega}_k \tag{1}$$

where $L(D_k, \boldsymbol{M}_k)$ is the task-specific loss (e.g., cross-entropy for classification) incurred on $D_k$ using the expertises in $\boldsymbol{M}'_k$ which comprises a new (learnable) expertise $\Delta\boldsymbol{\omega}_k$ and those retrieved from $\boldsymbol{M}_{k-1}$ using a retrieval procedure based on $\gamma_{\boldsymbol{\theta}}$ (e.g., top-$\kappa$ highest). For example, $\gamma_{\boldsymbol{\theta}}$ can be a learnable neural network (Wang et al., 2023a) or cosine distance (Wang et al., 2022b;c). The optimization task might also involve a (data-free) regularization term $R(\boldsymbol{M}'_k)$ that constrains the new expertise with respect to the (relevant) retrieved expertises to avoid knowledge interference. The retrieval and regularization loss terms are also weighted with scalar parameters $\lambda_1$ and $\lambda_2$. The database $\boldsymbol{M}_k = \boldsymbol{M}_{k-1} \cup \Delta\boldsymbol{\omega}_k$ is then updated with the new expertise $\Delta\boldsymbol{\omega}_k$.

**Existing Literature.** We summarize below two notable approaches in continual fine-tuning (CFT).

**1. Input-Adaptation CFTs.** These methods are based on prompt-tuning approaches (Lester et al., 2021), which model $\boldsymbol{M}_k$ as a set of task-specific prompts or prefix tokens to be appended to the input sequences. These prompts will be updated selectively based on their estimated relevance to the current task (Wang et al., 2022c). This requires continual updates to $\boldsymbol{\theta}$ to learn a retrieval score function $\gamma_{\boldsymbol{\theta}}$ that retrieves a subset of the most relevant prompts from $\boldsymbol{M}_k$ for each input $\boldsymbol{x}$ in the current task. More recent prompt-based CL methods (Wang et al., 2022b; Smith et al., 2023; Wang et al., 2022a; 2023a) have generalized the above approach with different regularization methods and organizations of the prompt parameters, including generating new prompts as the model observes more tasks (see Appendix I). However, continually updating the prompt retrieval function retains the risk of forgetting with diverse task sequences (see Section 5).

**2. Parameter-Adaptation CFTs.** These methods are based on fine-tuning algorithms that learn low-rank (LoRA) updates $\Delta\boldsymbol{\omega}_k = \boldsymbol{B}\boldsymbol{A}^\top$ which are added to the frozen weights of a pre-trained model (Hu et al., 2021) to augment its input representation. The matrices $\boldsymbol{B} \in \mathbb{R}^{p \times r}$ and $\boldsymbol{A} \in \mathbb{R}^{q \times r}$ are learnable and have low rank $r \ll \min(p, q)$. LoRA is useful for CL because it confines task-specific adaptation to a small set of parameters and anchors prior knowledge in the pre-trained backbone, allowing the pre-trained model to retent its prior knowledge. For general architectures, a pair of LoRA matrices $\boldsymbol{A}, \boldsymbol{B}$ is learned for each pre-trained weight matrix that needs fine-tuning. Like prompt-based CL, LoRA-based CL also maintains and grows a pool of adaptation parameters as the model observes more task data (McDonnell et al., 2023). However, since LoRA parameters do not live in the input embedding space, it is not trivial to define a similarity measure (as was done in prompt-based CL) to facilitate test-time retrieval of most relevant past updates.

To sidestep this and mitigate retriever forgetting, RanPAC McDonnell et al. (2023) uses LoRA parameters of the first encountered task $\Delta\boldsymbol{\omega}_1$ to generate input embeddings and update class embeddings for each subsequent task. This sufficient statistics enable a parameter-free classification scheme for any test input $\boldsymbol{x}_*$ using linear discriminant analysis (McDonnell et al., 2023). Other approaches such as InfLoRA (Liang & Li, 2024a) and SD-LoRA (Wu et al., 2025a) instead use the most recent update to generate embeddings. However, as the task diversity increases over time, this lack of test-time representation adaptation will degrade performance (see Fig. 3, Section 5).

**Overview of our proposed approach.** To facilitate both test-time adaptation and retriever forgetting mitigation, we develop a theoretical framework to analyze a broad class of parameter-free retrieval methods. This is done via learning and organizing low-complexity representation signature in a database. Our framework design is generic to the choice of of the signature function. As this design is parameter-free, it mitigates retriever forgetting. To gain insight into the retrieval error rate, we develop theoretical bounds linking it to the clustering structure of the signature patterns in the database. This is discussed in Section 3 and exploited to devise a practical algorithm in Section 4.

## 3 PARAMETER-FREE RETRIEVAL WITH PROVABLE ERROR RATE

This section studies a generic class of parameter-free retrieval algorithms for continual fine-tuning based on a concept of retrieval signature. This is intuitively a statistical descriptor associated with each task-specific adaptation module $\Delta\boldsymbol{\omega}$ (e.g., prompt or LoRA) that allows us to assess its similarity to a given test input $\boldsymbol{x}_*$. This is constructed by noting that each adaptation module $\Delta\boldsymbol{\omega} : \boldsymbol{x} \rightarrow \boldsymbol{h}$ modifies the pre-trained model in a task-specific manner which results in a distinct or signature distribution $D(\boldsymbol{h})$ over input embeddings $\boldsymbol{h}$ that reflects the characteristics of its corresponding task. The statistics of such signature distribution can then be used as a basis to guide retrieval as detailed in Section 3.1. The retrieval error rate can then be quantified explicitly based on the statistical properties of these signature distributions as further discussed in Section 3.2.

### 3.1 PARAMETER-FREE RETRIEVAL ALGORITHM

**I. Intuition.** Suppose an unseen input $\boldsymbol{x}_*$ belongs to a data regime of a specific task that is best embedded using a particular representation adaptation module $\Delta\boldsymbol{\omega}$, then its embedding $\boldsymbol{h}_*$ under $\Delta\boldsymbol{\omega}$ must have a high likelihood score $D(\boldsymbol{h}_*)$ according to $\Delta\boldsymbol{\omega}$'s signature distribution $D(.)$. Given a database KB containing (key, value) pairs $(\Delta\boldsymbol{\omega}, D)$, parameter-free retrieval can be achieved via:

$$\Delta\boldsymbol{\omega} \;\; = \;\; \underset{\Delta\boldsymbol{\omega}}{\arg\max}\, D\big(\Delta\boldsymbol{\omega}(\boldsymbol{x}_*)\big) \quad \text{over} \quad (\Delta\boldsymbol{\omega}, D) \,\in\, \text{KB} \,. \tag{2}$$

The core design of a parameter-free retrieval algorithm therefore centers on the adaptation module and how to organize its induced input embeddings into distinct signature distributions with specific likelihood functions. In this analysis, we will consider a class of clustering-based retrieval algorithms which are agnostic to the design of the adaptation module as discussed below.

**II. Design.** Let $\boldsymbol{x}$ be a test input, and $\boldsymbol{h}_k(\boldsymbol{x}) = h(\boldsymbol{x}; \Delta\boldsymbol{\omega}_k)$ be its induced embedding under $\Delta\boldsymbol{\omega}_k$. A matching signature for $\Delta\boldsymbol{\omega}_k$ could be characterized by a probabilistic model that summarizes its induced embeddings for inputs from $D_k$. One approach is to view the observed input embeddings $\boldsymbol{h}_k(\boldsymbol{x})$ where $\boldsymbol{x} \sim D_k$ as samples from a Gaussian distribution whose parameters, e.g., its mean and covariance matrices, can be learned via Maximum Likelihood Estimation (MLE):

$$(\boldsymbol{m}_k, \boldsymbol{\Lambda}_k) = \underset{\boldsymbol{m}, \boldsymbol{\Lambda}}{\arg\min} \sum_{\boldsymbol{x} \in D_k} S_k(\boldsymbol{x}; \boldsymbol{m}, \boldsymbol{\Lambda}) \;\triangleq\; \underset{\boldsymbol{m}, \boldsymbol{\Lambda}}{\arg\min} \sum_{\boldsymbol{x} \in D_k} \left(\boldsymbol{h}_k(\boldsymbol{x}) - \boldsymbol{m}\right)^{\top} \boldsymbol{\Lambda}^{-1} \left(\boldsymbol{h}_k(\boldsymbol{x}) - \boldsymbol{m}\right) . \tag{3}$$

The above signature function $S_k(\boldsymbol{x}; \boldsymbol{m}, \boldsymbol{\Lambda})$ is given by the negative log Gaussian likelihood of $\boldsymbol{h}_k(\boldsymbol{x})$ with mean $\boldsymbol{m}$ and covariance $\boldsymbol{\Lambda}$. During test time, each input $\boldsymbol{x}_*$ will be mapped to a nearest signature (and its corresponding training task ID) via finding $k_* = \arg\min_k S_k(\boldsymbol{x}_*)$.

To generalize the above to account for scenarios where the input embedding distribution can have multiple modes (whereas Gaussian is unimodal), we can instead learn a mixture of Gaussian components for the observed input embeddings. The learned mixture of Gaussian components in turn extend the task-specific single-key retrieval signature to a multi-key signature as defined below.

**Definition 3.1** (Multi-Key Signature). The multi-key signature of an adaptation module $\Delta\boldsymbol{\omega}_k$ learned for a task $D_k$ is a collection of $\tau$ Gaussian distributions with parameters $(\boldsymbol{m}_k^t, \boldsymbol{\Lambda}_k^t)_{t=1}^{\tau}$ such that the induced embedding $\boldsymbol{h}_k$ of each input $\boldsymbol{x} \sim D_k$ is distributed by $\mathbb{N}(\boldsymbol{m}_k^t, \boldsymbol{\Lambda}_k^t)$ for some

$t \in [\tau]$. The parameters $(\boldsymbol{m}_k^t, \boldsymbol{\Lambda}_k^t)_{t=1}^\tau$ are referred to as signature patterns while the corresponding distribution $\mathbb{N}(\boldsymbol{m}_k^t, \boldsymbol{\Lambda}_k^t)$ is referred to as signature distribution.

For each task and its corresponding adaptation module, the multi-key signature in Definition 3.1 can be learned by replacing the optimization task in Eq. 3 with:

$$\underset{\boldsymbol{m}^{1:\tau}, \boldsymbol{\Lambda}^{1:\tau}, \{\zeta_{it}\}}{\text{minimize}} \quad \sum_{t=1}^\tau \sum_{i=1}^{|D|} \zeta_{it} \cdot S_k\Big(\boldsymbol{x}_i; \boldsymbol{m}^t, \boldsymbol{\Lambda}^t\Big) \quad \text{such that} \quad \zeta_{it} \in \{0,1\} \quad \text{and} \quad \sum_{t=1}^\tau \zeta_{it} = 1 \ . \quad (4)$$

To avoid fixing the number of components $\tau$, we view the above as a clustering task and instead adopt a generalized non-parametric clustering scheme such as the Dirichlet Process for Gaussian Mixture Models (DP-GMMs) (Teh et al., 2010), which automatically learns the best value for $\tau$ in addition to the corresponding distribution parameters for each Gaussian component. During inference, each test input $\boldsymbol{x}_*$ is again matched to the training task with highest signature score, i.e., $(k_*, t_*) = \arg\min_{k,t} S_k(\boldsymbol{x}_*; \boldsymbol{m}_k^t, \boldsymbol{\Lambda}_k^t)$, which retrieves the specific $\Delta\boldsymbol{\omega}_k$ in our database.

## 3.2 PROVABLE RETRIEVAL ERROR RATE

This section establishes a direct link between the clustering structure of the aforementioned signature distributions and the retrieval error rate. As we assign each test input to the closest signature distribution under the corresponding representation adaptation module, retrieval error arises when an input $\boldsymbol{x}$ belongs to a task-specific regime with adaptation module $\Delta\boldsymbol{\omega}_k$ and signature component $\mathbb{N}(\boldsymbol{m}_k^t, \boldsymbol{\Lambda}_k^t)$ but is instead closer to another signature component $\mathbb{N}(\boldsymbol{m}_i^s, \boldsymbol{\Lambda}_i^s)$ of a different adaptation module $\Delta\boldsymbol{\omega}_i$ learned for another task. Such an error occurs when the two signature components are close to each other. Intuitively, a clustering structure with substantial overlap among clusters will induce high retrieval error, whereas well-separated clusters will reduce error. We will substantiate this intuition via formal concepts on clustered data distribution (see Assumption 3.2) and cluster separation factor (see Definition 3.3) as defined below.

**Assumption 3.2** (Clustered Data Distribution). Following our representation clustering algorithm in Eq. 4, we assume that for each task $k$ with a multi-key signature with $\tau$ components $(\boldsymbol{m}_k^t, \boldsymbol{\Lambda}_k^t)_{t=1}^\tau$, its data distribution $D_k$ can be decomposed into $\tau$ sub-distributions $\{D_k^t\}_{t=1}^\tau$ such that:

$$\boldsymbol{h}_k(\boldsymbol{x}) \ \sim \ \mathbb{N}\big(\boldsymbol{m}_k^t, \boldsymbol{\Lambda}_k^t\big) \quad \text{when} \quad \boldsymbol{x} \sim D_k^t(\boldsymbol{x}) \ . \quad (5)$$

This is a reasonable assumption as our clustering algorithm was developed to fit a $\tau$-component mixture of Gaussians $(\boldsymbol{m}_k^t, \boldsymbol{\Lambda}_k^t)_{t=1}^\tau$ to the input embeddings of each task $k$.

**Definition 3.3** (Cluster Separation Factor). The separation factor $\delta$ of a given sub-data distribution $D_k^t$ in Assumption 3.2 denotes how much the distance between an input $\boldsymbol{x} \sim D_k^t$ and a signature component $(\boldsymbol{m}_i^s, \boldsymbol{\Lambda}_i^s)$, measured under its corresponding adaptation module $\boldsymbol{h}_i(\boldsymbol{x}) = h(\boldsymbol{x}; \Delta\boldsymbol{\omega}_i)$, exceeds the embedding dimension $d$. That is,

$$\mathbb{E}_{\boldsymbol{x} \sim D_k^t}\Big[\big(\boldsymbol{h}_i(\boldsymbol{x}) - \boldsymbol{m}_i^s\big)^\top \big(\boldsymbol{\Lambda}_i^s\big)^{-1} \big(\boldsymbol{h}_i(\boldsymbol{x}) - \boldsymbol{m}_i^s\big)\Big] \ = \ \big(1 + \delta\big)d \ . \quad (6)$$

We can now bound the retrieval error rate in terms of the cluster separation factor $\delta$ below.

**Theorem 3.4.** *[Bounding Retrieval Error Rate] Suppose we are given $n$ training tasks with clustered data distributions $(D_1^t)_{t=1}^\tau, (D_2^t)_{t=1}^\tau, \ldots, (D_n^t)_{t=1}^\tau$. Let $\boldsymbol{E}_k^t$ denotes the event that a test input $\boldsymbol{x} \sim D_k^t$, where $D_k^t$ is the $t$-th sub-testing data distribution of task $k$, is not matched with the (correct) embedding function $\boldsymbol{h}_k(\boldsymbol{x})$. Under mild technical condition in A.1-A.2, we have*

$$\Pr\Big(\boldsymbol{E}_k^t\Big) \ \leq \ \exp\left(-\frac{d\delta^2}{4\delta + 16}\right) \ + \ (n-1)\tau \, \exp\left(-\frac{(d\delta/2 + \kappa)^2}{2\sigma^2 d + (2/3)(d\delta/2 + \kappa)}\right) \ with \quad (7)$$

$$\kappa \ \triangleq \ \min_{(i,s):i \neq k} \log \frac{|\boldsymbol{\Lambda}_i^s|}{|\boldsymbol{\Lambda}_k^t|} \quad and \quad \delta \ \geq \ \max\left(0, -\frac{2\kappa}{d}\right) \ . \quad (8)$$

*This means the error rate decreases exponentially in $\delta d \geq 0$ where $\kappa$ denotes the minimum difference between log-volume of false signature component $\mathbb{N}(\boldsymbol{m}_i^s, \boldsymbol{\Lambda}_i^s)$ and the true signature $\mathbb{N}(\boldsymbol{m}_k^t, \boldsymbol{\Lambda}_k^t)$.*

See Appendix A for the proof. The above bound implies that approximately the error rate decreases exponentially fast in $O(\delta d)$ but it also increases linearly in the number of task and representation

clusters $O(n\tau)$, i.e., $\Pr(\boldsymbol{E}_k^t) \lesssim O(n\tau) \exp(-O(\delta d))$. This intuitively means that a well-designed algorithm is needed to prevent the separation factor $\delta$ from becoming too small, which would otherwise negate the exponential decay in the error rate. Furthermore, we show that, theoretically, if $\delta$ is sufficiently large relative to the number of tasks and representation clusters, the retrieval error can be made arbitrarily small, despite its linear dependence on the number of tasks, shown in Theorem 3.5.

**Theorem 3.5.** *[Keeping Error Rate Arbitrarily Low] For $\epsilon > 0$, if $\delta$ is sufficiently large, $\Pr(\boldsymbol{E}_k^t) \leq \epsilon$. That is, we can choose $\delta$ as a function of $\epsilon$ to ensure $\Pr(\boldsymbol{E}_k^t) \leq \epsilon$, where:*

$$\delta \geq \frac{2}{d} \max \left\{ \frac{1}{4} \left( M + \left( M^2 - 24 d\sigma^2 \kappa \right)^{\frac{1}{2}} \right) - \kappa, \ \log \frac{N}{\epsilon} \left( 1 + \left( 1 + \frac{4d}{\log \frac{N}{\epsilon}} \right)^{\frac{1}{2}} \right) \right\}, \quad (9)$$

*where $\sigma^2$ is a constant factor computed in A.2 and $M = 3d\sigma^2 - \kappa$.*

See proof in Appendix A. Theorem 3.5 thus offers a valuable insight to design continual fine-tuning algorithm with parameter-free and provably low retrieval error rate. The key is to design the adaptation module and clustering structure that induce a large separation factor $\delta$ for each signature cluster.

Although optimizing this explicitly is difficult, it is intuitive that enforcing representation orthogonality across adaptation modules tends to enlarge cluster separation, ensuring low retrieval rate. We will develop a practical adaptive fine-tuning algorithm to achieve this in Section 4 and validate its effectiveness via extensive experiments on multiple benchmark in Section 5.

## 4 PRACTICAL ALGORITHM DESIGN

This section introduces an adaptive fine-tuning design based on low-rank adaptation (LoRA) (Hu et al., 2021). The key idea is to design a LoRA structure that incorporate knowledge transfer from old tasks while also discover orthogonal knowledge relevant to the new task. This will help improve the separation across their corresponding signature clusters as envisioned previously.

**1. Adaptation Module.** Let $\phi_{\tau,i}$ denote the $i^{\text{th}}$ new tuning direction for any task $\tau$. We note that each tuning direction $\phi_{\tau,i}$ represents a rank-1 update, and is parameterized as an outer product of two vectors in $\boldsymbol{b}_{\tau,i} \in \mathbb{R}^p$ and $\boldsymbol{a}_{\tau,i} \in \mathbb{R}^q$. For each new task $D_{k+1}$, we freeze old tuning directions, i.e., $\{\phi_{\tau,i} \mid 1 \leq \tau \leq k, 1 \leq i \leq r\}$, and subsequently represent its low-rank adaptation $\Delta\omega_{k+1}$ as a (learnable) linear combination of old (frozen) and $r$ new tuning directions $\{\phi_{k+1,i}\}_{i=1}^r$. That is:

$$\Delta\omega_{k+1} \triangleq \sum_{\tau=1}^k \sum_{i=1}^r s_{\tau,i} \phi_{\tau,i} + \sum_{i=1}^r \phi_{k+1,i} = \sum_{\tau=1}^k \boldsymbol{B}_\tau \boldsymbol{S}_\tau \boldsymbol{A}_\tau^\top + \boldsymbol{B}_{k+1} \boldsymbol{A}_{k+1}^\top, \quad (10)$$

where $\boldsymbol{A}_\tau \triangleq [\boldsymbol{a}_{\tau,1}, \dots, \boldsymbol{a}_{\tau,r}]$, $\boldsymbol{B}_\tau \triangleq [\boldsymbol{b}_{\tau,1}, \dots, \boldsymbol{b}_{\tau,r}]$, and $\boldsymbol{S}_\tau \triangleq \text{diag}[s_{\tau,1}, s_{\tau,2}, \dots, s_{\tau,r}]$ are diagonal matrices that contain learnable linear coefficients corresponding to old tuning directions. $\boldsymbol{B}_\tau \boldsymbol{S}_\tau \boldsymbol{A}_\tau^\top$ is therefore the knowledge transfer term which specifies the transfer from a previous task $\tau$ to task $k+1$, by mixing relevant tuning directions according to the signals encoded in $\boldsymbol{S}_\tau$

**2. Enforcing Representation Orthogonality.** Incorporating (frozen) prior tuning directions into the parameterization of $\Delta\omega_{k+1}$ enables knowledge transfer from previous tasks to new task. On the other hand, new tuning directions will be trained to capture unique adaptation patterns that are not relevant to $D_1, D_2, \dots, D_k$. To ensure these directions contribute genuinely novel information that are specific to $D_{k+1}$, they must be orthogonal to the existing ones. Otherwise, they would lie within the span of previously learned directions and offer no additional value to the model. This gives rise to a set of linear constraints $\langle \Delta\omega_{k+1}^\perp, \Delta\omega_\tau^\perp \rangle = 0$[1] for all $\tau \leq k$, where $\Delta\omega_\tau^\perp \triangleq \boldsymbol{B}_\tau \boldsymbol{A}_\tau^\top$.

We further note that, without any constraint on the knowledge transfer condition, $\boldsymbol{S}_\tau$ could assign non-zero weight to many directions from many past tasks even if they are only weakly relevant. To mitigate this and promote selective transfer, we additionally impose both $\ell_1$ and $\ell_2$-norm penalties than $\boldsymbol{S}_\tau$, which respectively encourage sparsity and low magnitude of the transfer. This forces the model to rely on a small number of past directions, and prevents excessive reliance on any single

---

[1] $\langle \boldsymbol{u}, \boldsymbol{v} \rangle \triangleq \text{vec}(\boldsymbol{u})^\top \text{vec}(\boldsymbol{v})$.

---

**Algorithm 1** PROTEUS Training

---

1: **INPUT:** pre-trained backbone $h(\boldsymbol{x}; \boldsymbol{\omega}_o)$ and stream of $m$ datasets $D_1, D_2, \ldots, D_m$
2: **OUTPUT:** database KB = {multi-key = $(\boldsymbol{\Lambda}_*^t, \boldsymbol{m}_*^t)_{t=1}^\mathcal{T}$, value = $\Delta\boldsymbol{\omega}_*$}; statistics $\boldsymbol{G}_S$, $(\boldsymbol{e}(c))_c$
3: initialize KB $\leftarrow \emptyset$, $\boldsymbol{G}_S \leftarrow \boldsymbol{0}$, $\boldsymbol{e}_S(c) \leftarrow \boldsymbol{0} \ \forall c$
4: **for** $k = 1$ **to** $m$ **do**
5:      compute **value** $= \Delta\boldsymbol{\omega}_k$ via solving Eq. 11
6:      compute **multi-key** $= (\boldsymbol{\Lambda}_k^t, \boldsymbol{m}_k^t)_{t=1}^\mathcal{T}$ via Eq. 4
7:      update KB $\leftarrow$ KB $+$ {**multi-key**, **value**}
8:      **for** $(\boldsymbol{x}, z) \sim D_k$ **do**
9:          $\boldsymbol{h}_S(\boldsymbol{x}) \triangleq \boldsymbol{h}_k(\boldsymbol{x}) = h(\boldsymbol{x}; \Delta\boldsymbol{\omega}_k)$ – see **II. Design** in Section 3.1
10:          $\boldsymbol{G}_S \leftarrow \boldsymbol{G}_S + \boldsymbol{h}_S(\boldsymbol{x})\boldsymbol{h}_S(\boldsymbol{x})^\top$
11:          $\boldsymbol{e}_S(z) \leftarrow \boldsymbol{e}_S(z) + m^{-1}|D_k|^{-1}\boldsymbol{h}_S(\boldsymbol{x})$
12:      **end for**
13: **end for**
14: **return** KB, $\boldsymbol{G}_S$, $(\boldsymbol{e}_S(c))_{c \,\in\, \text{class}}$

---

task. Hence, the continual learning loss at task $k + 1$ can be succinctly written as:

$$\underset{\Delta\boldsymbol{\omega}_{k+1}^\perp, \boldsymbol{S}}{\text{minimize}} \quad L\left(D_{k+1}, \sum_{\tau=1}^k \boldsymbol{B}_\tau \boldsymbol{S}_\tau \boldsymbol{A}_\tau^\top + \Delta\boldsymbol{\omega}_{k+1}^\perp\right) + \lambda\left(\alpha \cdot \|\boldsymbol{S}\|_1 + (1-\alpha) \cdot \|\boldsymbol{S}\|_2\right)$$

subject to $\quad \forall \tau \le k : \langle \Delta\boldsymbol{\omega}_{k+1}^\perp, \Delta\boldsymbol{\omega}_\tau^\perp \rangle = 0$, where $\quad \boldsymbol{S} = \text{blkdiag}[\boldsymbol{S}_1, \ldots, \boldsymbol{S}_k]$.      (11)

We also performed sensitivity analysis to demonstrate the relative importance of hyper-parameters $\lambda$ (see Section 5.3) and $\alpha$ (see Appendix L), and further provide empirical verification that the optimized solutions $\Delta\boldsymbol{\omega}_\tau^\perp$ indeed meet the orthogonality constraint (see Appendix F).

Most importantly, we show that our adaptive fine-tuning design yields sufficiently large cluster separation factors across signature clusters of the learned LoRA modules to guarantee a retrieval error rate below $4\%$. This is validated by comparing the empirical separation factor $\delta$ (see Eq. 6) with the theoretical $\delta$ required to achieve a target error rate $\epsilon$ (see Eq. 9). For each task, we construct signature distributions from input embeddings induced by LoRA units (see Eq. 11), compute the empirical $\delta$ (see Eq. 6), and evaluate the empirical retrieval error $\epsilon$ on the test set to obtain the minimum theoretical $\delta$ (see Eq. 9). Across all tasks, the empirical $\delta$ consistently exceeds the theoretical bound, demonstrating that our method enlarges cluster separation enough to ensure low retrieval error as predicted by the analysis in Section 3.2. This validation is visualized in Figure 2.

**3. Parameter-Free Prediction.** For test-time inference, we adopt the parameter-free LDA-based prediction scheme in (McDonnell et al., 2023) which is summarized below for clarity. Formally, let $\boldsymbol{h}_S(\boldsymbol{x}_*)$ is corresponding LoRA-augmented embedding for an unseen input $\boldsymbol{x}_*$ after running retrieval step to obtain $\boldsymbol{h}_S$, the final parameter-free prediction is defined as:

$$c_* = \arg\max_c \ \boldsymbol{h}_S(\boldsymbol{x}_*)^\top \left(\boldsymbol{G}_S + \gamma\boldsymbol{I}\right)^{-1} \boldsymbol{e}_S(c),$$      (12)

where $\gamma > 0$ is a user-specified noise level; $\boldsymbol{G}_S$ and $\boldsymbol{e}_S(c)$ are LDA statistics (see Algorithm 1). The overview of our PROTEUS framework is illustrated in Fig. 1. Its training pseudocode is provided in Alg. 1 and its inference pseudocode is deferred to Appendix B. We evaluate its (empirical) memory consumption in Fig. 8 and analyze its (theoretical) memory complexity in Appendix G.

## 5 EMPIRICAL STUDIES

### 5.1 EXPERIMENT SETUP

**1. Datasets.** To demonstrate the robustness of PROTEUS, we compare its performance with state-of-the-art CL baselines using pre-trained models across diverse benchmarks with varying task semantic gaps. We follow the setup in (Kim et al., 2024), simulating 3 gap scenarios, as described below.

▶ **Uniformly Mild.** We use the CIFAR-100 (Krizhevsky et al., 2009) (100 classes, 600 images each) and ImageNet-R (Hendrycks et al., 2021a) (200 classes, 100 images each) datasets. We partition ImageNet-R and CIFAR-100 into 10 task subsets with 20 and 10 classes each, respectively.

▶ **Uniformly Abrupt.** We experiment with the ImageNet-A dataset (Hendrycks et al., 2021b) and two versions of the Visual Task Adaptation Benchmark (VTAB) (Zhai et al., 2019), VTAB5T-Small and VTAB5T-Large. ImageNet-A comprises examples from 200 ImageNet classes that are commonly misclassified by a ResNet model trained on ImageNet. We uniformly split this dataset into 10 task subsets with 20 classes each. As the misclassification patterns across those examples are diverse, this simulation results in tasks with abrupt changes in semantic gap. The VTAB5T-Large dataset (Zhai et al., 2019) was created via sampling 5 most semantically distinct task subsets from the original VTAB19T dataset which contains 19 distinct visual datasets. The VTAB5T-Small dataset is a downsampled version of VTAB5T-Large (Zhou et al., 2023).

▶ **Varying.** We follow the setup in (Kim et al., 2024) to construct VTAB-Sim50 from VTAB by selecting 4 most distinct tasks and re-partitioning their data into new task subsets with 50% overlap.

**2. Baselines and Metrics.** We compare PROTEUS against prompt-based CL methods such as L2P (Wang et al., 2022c), AdaPromptCL (Kim et al., 2024), DualPrompt (Wang et al., 2022b), CODA-Prompt (Smith et al., 2023), SPrompt (Wang et al., 2022a), and HiDe-Prompt (Wang et al., 2023a); and LoRA-based CL methods such as InfLoRA (Liang & Li, 2024a), SD-LoRA (Wu et al., 2025a), and RanPAC (McDonnell et al., 2023). We measure and report the *average accuracy* (higher is better) $\mathrm{ACC}_\kappa = (1/\kappa) \sum_{\tau=1}^{\kappa} \mathrm{acc}(\tau, \kappa)$ where $\mathrm{acc}(\tau, \kappa)$ denotes model's prediction accuracy on task $\tau$ after learning task $\kappa$. See Appendix E for additional metrics and experiment details.

Table 1: Final average accuracy achieved by PROTEUS and other baselines across three semantic gap scenarios with diverse benchmark datasets. HiDE-Prompt appears to throw OOM issues or crash on the large scale datasets VTAB5T-large and VTAB5T-Sim50. The best and second-best values are highlighted in red and blue colors, respectively.

| Methods | Uniformly Mild | | Uniformly Abrupt | | | Varying |
|---|---|---|---|---|---|---|
| | CIFAR-100 | ImageNet-R | ImageNet-A | VTAB5T-large | VTAB5T-small | VTAB-Sim50 |
| L2P | $84.59_{\pm0.44}$ | $61.48_{\pm0.21}$ | $45.36_{\pm1.66}$ | $60.46_{\pm1.24}$ | $65.32_{\pm5.14}$ | $80.24_{\pm1.46}$ |
| DualPrompt | $85.26_{\pm0.26}$ | $69.38_{\pm0.26}$ | $49.37_{\pm0.79}$ | $71.84_{\pm0.37}$ | $75.30_{\pm1.57}$ | $87.74_{\pm0.33}$ |
| CODAPrompt | $84.24_{\pm3.11}$ | $75.39_{\pm0.48}$ | $52.86_{\pm0.41}$ | $25.57_{\pm1.14}$ | $73.52_{\pm0.27}$ | $86.01_{\pm1.91}$ |
| SPrompt | $87.04_{\pm0.34}$ | $67.71_{\pm0.73}$ | $40.49_{\pm0.26}$ | $77.37_{\pm0.67}$ | $85.64_{\pm0.28}$ | $90.69_{\pm0.22}$ |
| AdaPromptCL | $79.87_{\pm2.10}$ | $65.63_{\pm0.78}$ | $48.67_{\pm0.80}$ | $65.48_{\pm1.83}$ | $71.86_{\pm2.86}$ | $77.86_{\pm1.62}$ |
| HiDE-Prompt | $92.79_{\pm0.94}$ | $73.00_{\pm0.48}$ | $40.81_{\pm0.74}$ | OOM | $68.44_{\pm1.14}$ | OOM |
| InfLoRA | $86.45_{\pm0.21}$ | $74.55_{\pm0.50}$ | $47.88_{\pm0.71}$ | $32.58_{\pm1.81}$ | $90.55_{\pm0.88}$ | $91.25_{\pm0.25}$ |
| SD-LoRA | $87.08_{\pm0.44}$ | $75.81_{\pm0.67}$ | $52.03_{\pm0.64}$ | $26.13_{\pm5.59}$ | $91.02_{\pm0.76}$ | $84.91_{\pm1.95}$ |
| RanPAC | $92.01_{\pm0.19}$ | $78.08_{\pm0.30}$ | $62.76_{\pm1.09}$ | $85.84_{\pm0.24}$ | $92.34_{\pm0.13}$ | $90.22_{\pm0.24}$ |
| PROTEUS | $94.10_{\pm0.64}$ | $82.17_{\pm0.54}$ | $63.19_{\pm0.82}$ | $89.37_{\pm1.94}$ | $94.24_{\pm1.73}$ | $95.89_{\pm0.80}$ |

## 5.2 MAIN RESULTS

Table 1 shows that PROTEUS consistently outperforms all baselines across all semantic gap scenarios. In the *uniformly mild* scenario, PROTEUS significantly improves over prompt-based CL baselines up to 14.23% on CIFAR-100, and up to 20.69% on ImageNet-R. PROTEUS outperforms RanPAC (McDonnell et al., 2023) (current SOTA on LoRA-based CL) on both CIFAR-100 and ImageNet-R by substantial margins of 2.09% and 4.09%, respectively. In the more challenging *uniformly abrupt* scenario, our method still outperforms RanPAC.

Against the best prompt-based CL methods across all datasets (ImageNet-A, VTAB5T-Large, and VTAB5T-Small), PROTEUS consistently achieves 10% better performances. Finally, in the *varying* scenario, PROTEUS surpasses the best two baselines, InfLoRA and S-Prompt, by a substantial margin of 5%. We observe that LoRA-based CL methods (RanPAC and PROTEUS) generally outperforms prompt-based CL. This is reasonable given LoRA-based methods have more control over model adaptation as their LoRA units adapt pre-trained weights directly rather than indirectly via adding prompts to input sequences. Additional results on forgetting metric are also reported (see Appendix D), showing that PROTEUS attains the best top-1 average forgetting metric.

Fig. 3 shows the average accuracy of PROTEUS and other baselines across 10 CL iterations on three datasets. In the *uniformly abrupt* setting (VTAB-5T-Small), both PROTEUS and RanPAC outperform other continual learning methods across most iterations, demonstrating consistently strong performances. In the *uniformly mild* (CIFAR-100) and *varying* (VTAB-Sim50) settings, both InfLoRA

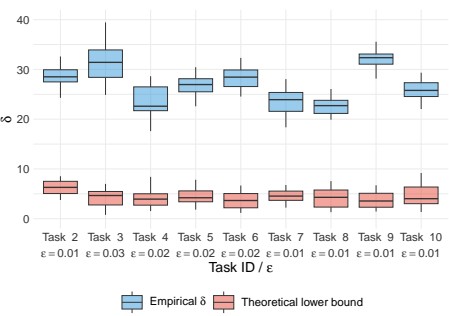

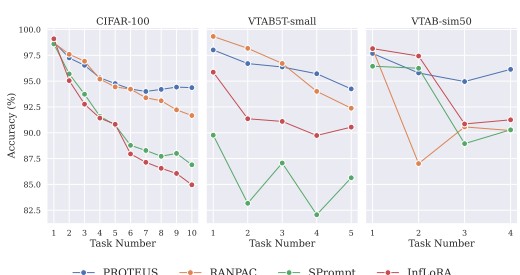

Figure 2: Box plots showing the distribution of our empirical cluster separation factor $\delta$ clearly lies above the distribution of its minimum requirement to guarantee low retrieval error rate $\epsilon$ across tasks in the Split CIFAR-100 benchmark.

Figure 3: PROTEUS has the most stable performance and highest final accuracies on 3 datasets: CIFAR-100, VTAB5T-small and VTAB-sim50.

Table 2: Performance achieved by PROTEUS with and without retrieval across various scenarios and datasets.

| CL Datasets | w/ Retrieval | w/o Retrieval |
|---|---|---|
| CIFAR-100 | **94.10** | 76.96 |
| ImageNet-R | **82.17** | 57.07 |
| ImageNet-A | **63.19** | 35.03 |
| VTAB5T-large | **89.37** | 70.07 |
| VTAB5T-small | **94.24** | 61.65 |
| VTAB-Sim50 | **95.89** | 78.90 |

Table 3: Retrieval accuracy achieved by retrieval-based CL methods on ImageNet-R, VTAB5T-small and VTAB5T-Sim50. Retrieval accuracy is defined as the percentage of instances where the correct task identity is successfully retrieved.

| Methods | ImageNet-R | VTAB-small | VTAB-Sim50 |
|---|---|---|---|
| D-Prompt | 38.59 | 61.35 | 82.40 |
| SPrompt | 42.52 | 77.26 | 90.43 |
| HiDE | 47.23 | 69.87 | OOM |
| PROTEUS | **96.02** | **97.97** | **97.16** |

and RanPAC achieve slightly higher accuracy on early tasks but experience a rapid decline in subsequent tasks, resulting in lower overall accuracy compared to PROTEUS. This behavior is expected since RanPAC only uses the first learned LoRA unit to perform prediction, and thus struggles to adapt as the model observes more tasks. On the other hand, InfLoRA always uses the most recent update, losing task-specific optimal embeddings acquired earlier on during training. In contrast, PROTEUS's adaptive fine-tuning component demonstrates a robust ability to safeguard against task interference, which leads to superior performance. Prompt-based CL methods again show a sharper performance decline and lower retrieval accuracy (Fig. 3, Table 3), highlighting their susceptibility to forgetting due to parameterized retrieval, especially with large task semantic gaps.

## 5.3 ABLATION STUDIES

**Impact of task retrieval on performance.** We compare PROTEUS with a variant that uses the LoRA unit $\Delta\omega_m$ of the last task $D_m$ which accummulates knowledge transfer from all previous tasks. We report this result in Table 2, which shows a significant performance drop (17%-32%) when task retrieval is removed from PROTEUS. The largest drop occurs on the uniformly abrupt scenario. This underscores the importance of task retrieval. Additionally, compared to the best baselines among retrieval-based methods with different task retrieval designs, PROTEUS achieves up to 58% improvement in retrieval accuracy (see Table 3), hence corroborating our previous observation.

**Adaptive Fine-Tuning Enhances Retrieval.** To demonstrate the synergies between the adaptive fine-tuning and task retrieval components in Section 4, we compare PROTEUS (with learned $\boldsymbol{S}_\tau = \boldsymbol{S}_*$) against a variant that fixes $\boldsymbol{S}_\tau = 0$ for all task $\tau$, and drops the constraint in Eq. 11 (i.e., no knowledge transfer). The result in Fig. 4 reveals that without knowledge transfer, the retrieval accuracy drops significantly and consistently across various task scenarios. This is expected since the LoRA units learned without regard to previous experience might overlap in their tuning directions, and thus encodes less useful information patterns. Table 10 compares these baselines against the

heuristic scenario where all previous tasks contribute equally to learning the current one ($S_\tau = I$), showing an even worse performance degradation if we rely excessively on past knowledge.

Additional ablation can be found in Appendix F, where we validate the orthogonality of LoRA units, visualize the sparsity of the selection matrix and study the effect of selecting Top-K Gaussians.

## 6 CONCLUSION

Our work provides a fresh perspective on continual fine-tuning by establishing a principled connection between task-retrieval error and the clustering properties on a space of representation signature. We develop a new theoretical framework with rigorous guarantees that characterize when retrieval can remain reliable under evolving task distributions. This is comprehensively validated by our extensive experiments across diverse benchmarks, which demonstrate that the proposed framework not only improves retrieval accuracy but also achieves strong predictive performance while mitigating forgetting. These results show that our approach effectively unifies the strengths of input- and parameter-adaptation while avoiding their respective limitations. Integrating theoretical rigor with comprehensive empirical evidence, this work advances a new paradigm for continual fine-tuning that emphasizes parameter-free retrieval grounded in structural representation properties. We believe this opens a promising direction for building provably effective continual learning systems.

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

# APPENDIX

**Code Release.** Our experimental code is released and maintained at `https://anonymous.4open.science/r/PROTEUS-4637`.

# A    BOUNDING THE TASK-RETRIEVAL ERROR RATE

This section provides a theoretical analysis regarding the error rate of our (parameter-free) task-retrieval procedure in Section 3.1 under mild assumptions on the learned clustering of the learned representation across tasks. This is formalized below.

## A.1    LISTS OF ASSUMPTIONS AND LAYMEN EXPLANATIONS

**Assumption A.1** (Clustered Data Distribution). Following our representation clustering algorithm in Eq. 4, we assume that for each task $k$ with a multi-key signature with $\tau$ components $(\boldsymbol{m}_k^t, \boldsymbol{\Lambda}_k^t)_{t=1}^\tau$, its data distribution $D_k$ can be decomposed into $\tau$ sub-distributions $\{D_k^t\}_{t=1}^\tau$ such that:

$$\boldsymbol{h}_k(\boldsymbol{x}) \sim \mathbb{N}\big(\boldsymbol{m}_k^t, \boldsymbol{\Lambda}_k^t\big) \quad \text{when} \quad \boldsymbol{x} \sim D_k^t(\boldsymbol{x}) . \tag{13}$$

This is a reasonable assumption as our clustering algorithm was developed to fit a $\tau$-component mixture of Gaussians $(\boldsymbol{m}_k^t, \boldsymbol{\Lambda}_k^t)_{t=1}^\tau$ to the input embeddings of each task $k$.

**Assumption A.2** (Cluster Separation). We also assume the following mild regularity condition between the representation clusters:

**A2-1.** For a representation cluster $(\boldsymbol{m}_i^s, \boldsymbol{\Lambda}_i^s)$, the largest (geometric) spread of its uncertainty volume is not exponentially larger, relative to the representation dimension $d$, than that of $k$-th task's representation cluster $(\boldsymbol{m}_k^t, \boldsymbol{\Lambda}_k^t)$. That is,

$$\max_{(i,s): i \neq k} \big|\boldsymbol{\Lambda}_i^s\big| \quad \leq \quad e^{d\sigma^2} \big|\boldsymbol{\Lambda}_k^t\big| \tag{14}$$

**A2-2.** The average distance between an input sampled from $D_k^t$ and the mean of a different representation cluster $(\boldsymbol{m}_i^s, \boldsymbol{\Lambda}_i^s)$, measured under the representation function $\boldsymbol{h}_i(\boldsymbol{x})$ associated with that cluster, exceeds $d$ by a multiplicative or *separation factor* $1 + \delta$. That is,

$$\mathbb{E}_{\boldsymbol{x} \sim D_k^t} \left[ \big(\boldsymbol{h}_i(\boldsymbol{x}) - \boldsymbol{m}_i^s\big)^\top \big(\boldsymbol{\Lambda}_i^s\big)^{-1} \big(\boldsymbol{h}_i(\boldsymbol{x}) - \boldsymbol{m}_i^s\big) \right] \quad = \quad \big(1 + \delta\big)d . \tag{15}$$

**A2-3.** The representation variance of an input $\boldsymbol{x}$ sampled from $D_k^t$, measured under the representation function $\boldsymbol{h}_i(\boldsymbol{x}) \in \mathbb{R}^d$ of another task $i \neq k$ is bounded by $O(d)$. That is, $\mathbb{V}_{\boldsymbol{x} \in D_k^t}[\boldsymbol{h}_i(\boldsymbol{x})] \leq d\sigma^2$.

**Intuition.** All these assumptions are reasonable regularity conditions meant to *exclude cases with irregular representation clustering structures*, such as exponentially differences in uncertainty volumes across clusters (**A2-1**), irregularly close to incorrect cluster assignment (**A2-2**), excessively large representation variance (**A2-3**), which are unlikely to occur under DP-GMM.

## A.2    THEORETICAL ANALYSIS

Under these assumptions, we will establish the following key results, which are first stated in high-level language and then proved mathematically.

**I. High-Level Results.** Our key results are two-fold:

**R1.** The retrieval error rate of our algorithm (PROTEUS) during inference decays exponentially fast at a rate measured by a rational function of the input dimension $d$, separation factor $\delta$, and variance factor $\sigma^2$. It however grows linearly in the number of tasks (Theorem 3.4).

**R2.** It is theoretically possible to offset this linear growth and drive the error arbitrarily close to zero by making the separation factor sufficiently large (Theorem 3.5).

Intuitively, this is achieved by enforcing orthogonality among fine-tuning directions and non-parametrically increasing the number of clusters when appropriate, which together minimize cluster overlap (i.e., increase separation) and reduce both cluster spread and representation variance.

**Empirical Verification.** This is empirically validated in Fig. 4, which shows: (1) near-optimal retrieval accuracy when adaptive fine-tuning enforces orthogonality, even as the number of tasks increases; and (2) a sharp degradation in retrieval accuracy when such adaptive tuning is absent.

**II. Technical Proofs.** We now substantiate the above high-level results with mathematical proofs.

**Theorem 3.4.** *[Bounding Retrieval Error Rate] Suppose we are given $n$ training tasks with clustered data distributions $(D_1^t)_{t=1}^\tau, (D_2^t)_{t=1}^\tau, \ldots, (D_n^t)_{t=1}^\tau$. Let $\boldsymbol{E}_k^t$ denotes the event that a test input $\boldsymbol{x} \sim D_k^t$, where $D_k^t$ is the $t$-th sub-testing data distribution of task $k$, is not matched with the (correct) embedding function $\boldsymbol{h}_k(\boldsymbol{x})$. Under mild technical condition in A.1-A.2, we have*

$$\Pr\left(\boldsymbol{E}_k^t\right) \leq \exp\left(-\frac{d\delta^2}{4\delta + 16}\right) + (n-1)\tau \exp\left(-\frac{(d\delta/2 + \kappa)^2}{2\sigma^2 d + (2/3)(d\delta/2 + \kappa)}\right) \quad with \quad (7)$$

$$\kappa \triangleq \min_{(i,s):i\neq k} \log\frac{|\boldsymbol{\Lambda}_i^s|}{|\boldsymbol{\Lambda}_k^t|} \quad and \quad \delta \geq \max\left(0, -\frac{2\kappa}{d}\right). \quad (8)$$

*This means the error rate decreases exponentially in $\delta d \geq 0$ where $\kappa$ denotes the minimum difference between log-volume of false signature component $\mathbb{N}(\boldsymbol{m}_i^s, \boldsymbol{\Lambda}_i^s)$ and the true signature $\mathbb{N}(\boldsymbol{m}_k^t, \boldsymbol{\Lambda}_k^t)$.*

**Proof.** To avoid cluttering the notations, we assume the same number $\tau$ of representation clusters per task in what follows. The proof can be trivially extended to cases with different $\tau$ per task.

Under Assumption A.1, the embedding function $h_k(\boldsymbol{x})$ transforms each sub-distribution $D_k^t$ of task $k$ into a Gaussian $\mathbb{N}(\boldsymbol{m}_k^t, \boldsymbol{\Lambda}_k^t)$. Hence, when $\boldsymbol{E}_k^t$ occurs, there must exist $(i,s) : i \neq k$ for which,

$$\log \mathbb{N}(h_i(\boldsymbol{x}) \mid \boldsymbol{m}_i^s, \boldsymbol{\Lambda}_i^s) > \log \mathbb{N}(h_k(\boldsymbol{x}) \mid \boldsymbol{m}_k^t, \boldsymbol{\Lambda}_k^t) \Rightarrow Z_k^t > Z_i^s + \log\frac{|\boldsymbol{\Lambda}_i^s|}{|\boldsymbol{\Lambda}_k^t|}. \quad (16)$$

where $Z_k^t = (h_k(\boldsymbol{x}) - \boldsymbol{m}_k^t)^\top (\boldsymbol{\Lambda}_k^t)^{-1}(h_k(\boldsymbol{x}) - \boldsymbol{m}_k^t)$, $Z_i^s = (h_i(\boldsymbol{x}) - \boldsymbol{m}_i^s)^\top (\boldsymbol{\Lambda}_i^s)^{-1}(h_i(\boldsymbol{x}) - \boldsymbol{m}_i^s)$.

This consequently implies

$$\boldsymbol{E}_k^t \Rightarrow Z_k^t > \min_{(i,s):i\neq k}\left(Z_i^s + \log\frac{|\boldsymbol{\Lambda}_i^s|}{|\boldsymbol{\Lambda}_k^t|}\right), \quad (17)$$

$$\Rightarrow Z_k^t > \min_{(i,s):i\neq k}\left(Z_i^s + \kappa\right) \equiv \boldsymbol{Q}_k^t. \quad (18)$$

Hence, $P(\boldsymbol{E}_k^t) \leq P(\boldsymbol{Q}_k^t)$. Following this, we further note that for any choice of $\theta$,

$$\Pr\left(\boldsymbol{E}_k^t\right) \leq \Pr\left(\boldsymbol{Q}_k^t\right) \leq \Pr\left(Z_k^t > \theta\right) + \sum_{i\neq k}\sum_{s=1}^\tau \Pr\left(Z_i^s < \theta - \kappa\right), \quad (19)$$

which holds due to the union bound. Now, by definition, $Z_k^t$ is a chi-square random variable with $d$ degrees of freedom. That is, $Z_k^t \sim \chi^2(d)$. Thus, if $\theta > d$, we can use the right-tail bound of $\chi^2(d)$ to measure $\Pr(Z_k^t > \theta)$ in the above,

$$\Pr\left(Z_k^t > \theta\right) \leq \exp\left(-\frac{(\theta - d)^2}{4d + 2(\theta - d)}\right) \quad when \quad \theta \geq d. \quad (20)$$

On the other hand, leveraging Assumption A.2 on the conditions of the mean and variance of $Z_i^s$, we can bound its left tail CDF with Bernstein's inequality,

$$\Pr\left(Z_i^s < \theta - \kappa\right) \leq \exp\left(-\frac{(\mathbb{E}[Z_i^s] - \theta + \kappa)^2}{2\mathbb{V}[Z_i^s] + (2/3)(\mathbb{E}[Z_i^s] - \theta + \kappa)}\right) \quad when \ \theta - \kappa \leq \mathbb{E}[Z_i^s], \quad (21)$$

where the expectation and variance are over $\boldsymbol{x} \sim D_k^t$. Under Assumption A.2-2, $\mathbb{E}[Z_i^s] = (1+\delta)d$. Thus, it is sufficient to choose $\theta - \kappa \leq (1+\delta)d$ or $\theta \leq (1+\delta)d + \kappa$ so that Eq. equation 21 holds. Plugging Eq. equation 20 and Eq. equation 21 into Eq. equation 19, we have:

$$\Pr\left(\boldsymbol{E}_k^t\right) \leq \exp\left(-\frac{(\theta - d)^2}{4d + 2(\theta - d)}\right) + (n-1)\tau \exp\left(-\frac{(\mathbb{E}[Z_i^s] - \theta + \kappa)^2}{2\mathbb{V}[Z_i^s] + (2/3)(\mathbb{E}[Z_i^s] - \theta + \kappa)}\right), \quad (22)$$

when $d \leq \theta \leq (1+\delta)d + \kappa$. Now, plugging in the facts that $\mathbb{E}[Z_i^s] = (1+\delta)d$ and $\mathbb{V}[Z_i^s] \leq \sigma^2 d$, we can slightly loosen (increase) the RHS of Eq. 22 to arrive at a cleaner bound,

$$\Pr\left(\boldsymbol{E}_k^t\right) \leq \exp\left(-\frac{(\theta - d)^2}{4d + 2(\theta - d)}\right) + (n-1)\tau \exp\left(-\frac{(d(1+\delta) - \theta + \kappa)^2}{2\sigma^2 d + (2/3)(d(1+\delta) - \theta + \kappa)}\right) \quad (23)$$

We can now choose $\theta = (1 + \delta/2)d$ which satisfies $d \leq \theta \leq (1 + \delta)d + \kappa$ given the premise $\delta \geq \max(0, -2\kappa/d)$ in Eq. 8. With this choice, Eq. 23 can be further simplified as

$$\Pr\left(\boldsymbol{E}_k^t\right) \leq \exp\left(-\frac{d\delta^2}{4\delta + 16}\right) + (n-1)\tau \exp\left(-\frac{(d\delta/2 + \kappa)^2}{2\sigma^2 d + (2/3)(d\delta/2 + \kappa)}\right) . \tag{24}$$

This completes our proof for Theorem 3.4. $\square$

**Remark 1.** The above bound implies that approximately the error rate decreases exponentially fast in $O(\delta d)$ but it also increases linearly in the number of task and representation clusters $O(n\tau)$, i.e., $\Pr(\boldsymbol{E}_k^t) \lesssim O(n\tau) \exp(-O(\delta d))$. This intuitively means that a well-designed algorithm is needed to prevent the separation factor $\delta$ from becoming too small, which would otherwise negate the exponential decay in the error rate. Furthermore, we show that, theoretically, if $\delta$ is sufficiently large relative to the number of tasks and representation clusters, the retrieval error can be made arbitrarily small, despite its linear dependence on the number of tasks.

**Theorem 3.5.** *[Keeping Error Rate Arbitrarily Low] For $\epsilon > 0$, if $\delta$ is sufficiently large, $\Pr\left(\boldsymbol{E}_k^t\right) \leq \epsilon$. That is, we can choose $\delta$ as a function of $\epsilon$ to ensure $\Pr\left(\boldsymbol{E}_k^t\right) \leq \epsilon$, where:*

$$\delta \geq \frac{2}{d} \max\left\{ \frac{1}{4}\left( M + \left(M^2 - 24d\sigma^2\kappa\right)^{\frac{1}{2}}\right) - \kappa, \ \log\frac{N}{\epsilon}\left(1 + \left(1 + \frac{4d}{\log\frac{N}{\epsilon}}\right)^{\frac{1}{2}}\right)\right\}, \tag{9}$$

*where $\sigma^2$ is a constant factor computed in A.2 and $M = 3d\sigma^2 - \kappa$.*

**Proof.** We will first show that with a sufficiently large $\delta$, it will occur that:

$$\exp\left(-\frac{d\delta^2}{4\delta + 16}\right) \geq \exp\left(-\frac{(d\delta/2 + \kappa)^2}{(2\sigma^2 d + (2/3)(d\delta/2 + \kappa))}\right) . \tag{25}$$

To find $\delta$ that satisfies Eq. 25, we note that $\exp(-d\delta^2/(4\delta + 16)) > \exp(-d\delta^2/(4\delta)) = \exp(-d\delta/4)$. Thus, it suffices to find $\delta$ such that

$$\exp\left(-\frac{d\delta}{4}\right) \geq \exp\left(-\frac{(d\delta/2 + \kappa)^2}{(2\sigma^2 d + (2/3)(d\delta/2 + \kappa))}\right) . \tag{26}$$

This is equivalent to finding $\delta$ such that

$$2z^2 - 3z\left(d\sigma^2 - \kappa/3\right) + 3\kappa d\sigma^2 \geq 0 , \tag{27}$$

where $z = d\delta/2 + \kappa$. Solving for $z$ and substituing in $\delta = (2/d)(z - \kappa)$ reveals the condition,

$$\delta \geq \frac{2}{d}\left[\frac{1}{4}\left(3d\sigma^2 - \kappa + ((3d\sigma^2 - \kappa)^2 - 24d\sigma^2\kappa)^{\frac{1}{2}}\right) - \kappa\right] . \tag{28}$$

Under Eq. 28 as well as the condition on $\delta$ in Theorem 3.4, Eq. 25 holds, which consequently implies

$$\Pr\left(\boldsymbol{E}_k^t\right) \leq N \exp\left(-\frac{d\delta^2}{4\delta + 16}\right) , \tag{29}$$

where $N = (n-1)\tau + 1$. We can continue to solve for $\delta$ such that $N \exp(-d\delta^2/(4\delta + 16)) \leq \epsilon$, which guarantees $\Pr(\boldsymbol{E}_k^t) \leq \epsilon$. Solving for $\delta$ now requires solving another quadractic inequality,

$$d\delta^2 - 4\delta \log\frac{N}{\epsilon} - 16\log\frac{N}{\epsilon} \geq 0 . \tag{30}$$

This implies $\delta \geq (2/d)(\log(N/\epsilon) + (\log(N/\epsilon)(\log(N/\epsilon) + 4d))^{1/2})$. Combining this with previous constraints on $\delta$, we obtain the final choice of $\delta$,

$$\delta \geq \frac{2}{d} \max\left\{ \frac{1}{4}\left( M + \left(M^2 - 24d\sigma^2\kappa\right)^{\frac{1}{2}}\right) - \kappa, \ \log\frac{N}{\epsilon}\left(1 + \left(1 + \frac{4d}{\log\frac{N}{\epsilon}}\right)^{\frac{1}{2}}\right)\right\}, \tag{31}$$

where $M = 3d\sigma^2 - \kappa$. This will make $\Pr(\boldsymbol{E}_k^t) \leq \epsilon$ for any arbitrarily small value of $\epsilon$ as desired, provided that we have an algorithm design that can learn the representation cluster with the above separation factor in Eq. 31. $\square$

---

**Algorithm 2** PROTEUS Inference

---

1: **INPUT:** pre-trained backbone $h(\boldsymbol{x}; \boldsymbol{\omega}_o)$, test input $\boldsymbol{x}_*$, LDA parameters $\boldsymbol{G}$ and $(\boldsymbol{e}(c))_c$, fine-tuning database KB – see line 7 of Algorithm 1.
2: **OUTPUT:** class label $c_*$
3: retrieve $(\boldsymbol{\Lambda}_{k_*}^{t_*}, \boldsymbol{m}_{k_*}^{t_*})$ for $\boldsymbol{x}_*$ – see lines 200-201 in main text
4: retrieve $\Delta\boldsymbol{\omega}_S(\boldsymbol{x}_*) = \text{KB}[(\boldsymbol{\Lambda}_{k_*}^{t_*}, \boldsymbol{m}_{k_*}^{t_*})]$ – see line 7 in Algorithm 1
5: compute prediction $c_*$ via Eq. 12
6: **return** $c_*$

---

## B    Inference Algorithm of PROTEUS

## C    Complexity Analysis of PROTEUS

We analyze the computational complexity of PROTEUS Training and PROTEUS Inference in Algorithm 1 and 2. Let $m$ denote the total number of tasks, $T$ denotes the maximum number of Gaussian components and $\mathcal{C}$ be the number of classes.

### C.1    Complexity of PROTEUS Training

The main loop (line 4-13, Alg. 1) is executed once for each $m$ tasks, resulting in a complexity that grows linearly in $m$. We analyze the complexity for each task $k$ below.

1. Computing the **value** $\Delta\boldsymbol{\omega}_k$ via solving Eq. 11 in line 5 takes $\mathcal{O}(C(\Delta\boldsymbol{\omega}_k))$, where $C(\Delta\boldsymbol{\omega}_k)$ is the cost of the LoRA fine-tuning.

2. Next, we compute the **multi-key** $= (\boldsymbol{\Lambda}_k^t, \boldsymbol{m}_k^t)_{t=1}^\tau$ for $\tau$ components via Eq. 4 –line 6. Let $T \geq \tau$ be the maximum number of Gaussian mixture components, the cost of learning this DP-GMM is upper-bounded by $\mathcal{O}(|\mathcal{D}_k|Td^2)$.

3. The inner loop in line 8-12 runs for $|\mathcal{D}_k|$ iterations. For each iteration, updating $\boldsymbol{G}_S + \boldsymbol{h}_S(\boldsymbol{x})\boldsymbol{h}_S(\boldsymbol{x})^\top$ (line 10) and $\boldsymbol{e}_S(z) \leftarrow \boldsymbol{e}_S(z) + m^{-1}|D_k|^{-1}\boldsymbol{h}_S(\boldsymbol{x})$(line 11) takes $\mathcal{O}(d^2)$ and $\mathcal{O}(d)$ time respectively. Thus, the total cost of the inner loop is $\mathcal{O}\left(|D_k|(d^2 + d)\right)$.

Finally, the computational complexity of PROTEUS training is:

$$\mathcal{O}\left(m\Big(C(\Delta\boldsymbol{\omega}_k) + |D_k|Td^2 + |D_k|(d^2 + d)\Big)\right) \quad = \quad \mathcal{O}\left(m\Big(C(\Delta\boldsymbol{\omega}_k) + Td^2|D_k|\Big)\right). \tag{32}$$

### C.2    Complexity of PROTEUS Inference

For each test input $\mathbf{x}_*$, we analyze the cost of inference in Alg. 2 below:

1. Retrieving $(\boldsymbol{\Lambda}_{k_*}^{t_*}, \boldsymbol{m}_{k_*}^{t_*})$ for $\boldsymbol{x}_*$ (line 3) takes $\mathcal{O}(Tmd^3)$.

2. Retrieving $\Delta\boldsymbol{\omega}_S(\boldsymbol{x}_*) = \text{KB}[(\boldsymbol{\Lambda}_{k_*}^{t*}, \boldsymbol{m}_{k_*}^{t*})]$ (line 4) takes $\mathcal{O}(1)$ time.

3. Computing $c_*$ via Eq. 12 (line 5) takes $\mathcal{O}(\mathcal{C}(d^3 + d^2 + d)) = \mathcal{O}(\mathcal{C}d^3)$.

The computational complexity of PROTEUS Inference is therefore $\mathcal{O}(Tmd^3 + \mathcal{C}d^3)$.

For example, fixing $T = 100$ (the largest $T$ in all settings), we report below the per-batch overhead (in seconds) on ImageNet-R as a function of the number of tasks $m$ in Table 4:

In the context of long sequences of tasks, we further run PROTEUS on CIFAR-100 with $m = 50$ tasks, 2 classes each task. The overhead (in seconds) each batch (batch size = 32) is a function of the no. of tasks $m$ as Table 5 below:

Table 4: Inference latency report on ImageNet-R per-batch (in seconds) as a function of the number of tasks $m$.

| Task 1 | Task 2 | Task 3 | Task 4 | Task 5 | Task 6 | Task 7 | Task 8 | Task 9 | Task 10 |
|--------|--------|--------|--------|--------|--------|--------|--------|--------|---------|
| 0.41 | 0.77 | 1.03 | 1.31 | 1.66 | 2.13 | 2.42 | 2.60 | 2.97 | 3.29 |

Table 5: Inference latency report on CIFAR-100 per-batch (in seconds) as a function of the number of tasks $m$.

| Ta. 2 | Ta. 5 | Ta. 10 | Ta. 15 | Ta. 20 | Ta. 25 | Ta. 30 | Ta. 35 | Ta. 40 | Ta. 45 | Ta. 50 |
|-------|-------|--------|--------|--------|--------|--------|--------|--------|--------|--------|
| 0.27 | 0.64 | 1.33 | 2.08 | 2.93 | 3.86 | 4.89 | 5.91 | 7.18 | 8.26 | 9.59 |

## D  ADDITIONAL RESULTS ON THE FORGETTING METRIC

Table 6 shows that PROTEUS achieves the lowest average forgetting rank among all baselines and consistently ranks within the top three best-performing baselines. In the *uniformly mild* and *varying* scenarios, PROTEUS ranks in the top two, with minimal gaps to the top performer (0.01 on CIFAR-100 and 0.36 on VTAB-Sim50, respectively). In the more challenging *uniformly abrupt* scenario, our method ranks within the top three, resulting in the lowest average forgetting rank across all three scenarios. This demonstrates that PROTEUS mitigates catastrophic forgetting more effectively than baselines across diverse task-semantic scenarios.

Table 6: Final average forgetting achieved by PROTEUS and other baselines across three semantic gap scenarios with diverse benchmark datasets.

| Methods | Uniformly Mild | | Uniformly Abrupt | | | Varying | Average Rank |
|---------|---------|---------|---------|---------|---------|---------|---------|
| | CIFAR-100 | ImageNet-R | ImageNet-A | VTAB5T-large | VTAB5T-small | VTAB-Sim50 | |
| L2P | $6.22_{\pm 0.16}$ (6) | $7.93_{\pm 0.66}$ (6) | $6.29_{\pm 0.33}$ (4) | $27.90_{\pm 2.14}$ (6) | $12.30_{\pm 3.01}$ (6) | $17.73_{\pm 1.69}$ (6) | 5.67 (6) |
| DualPrompt | $6.14_{\pm 0.59}$ (5) | $5.39_{\pm 0.19}$ (5) | $9.98_{\pm 0.16}$ (6) | $10.55_{\pm 2.27}$ (4) | $9.17_{\pm 2.57}$ (5) | $8.74_{\pm 0.80}$ (4) | 4.83 (5) |
| CODAPrompt | $2.01_{\pm 0.72}$ (1) | $1.61_{\pm 0.08}$ (2) | $4.73_{\pm 1.17}$ (2) | $25.70_{\pm 1.58}$ (5) | $6.55_{\pm 1.32}$ (4) | $9.67_{\pm 2.09}$ (5) | 3.17 (4) |
| SPrompt | $5.33_{\pm 0.23}$ (4) | $3.32_{\pm 0.08}$ (3) | $3.45_{\pm 0.29}$ (1) | $3.99_{\pm 1.19}$ (2) | $6.24_{\pm 0.39}$ (3) | $3.86_{\pm 0.76}$ (3) | 2.67 (3) |
| RanPAC | $2.87_{\pm 0.11}$ (3) | $4.55_{\pm 0.55}$ (4) | $7.02_{\pm 0.95}$ (5) | $0.31_{\pm 0.02}$ (1) | $1.85_{\pm 0.19}$ (1) | $1.05_{\pm 0.10}$ (1) | 2.50 (2) |
| PROTEUS | $2.02_{\pm 0.27}$ (2) | $1.45_{\pm 0.36}$ (1) | $5.35_{\pm 1.40}$ (3) | $6.54_{\pm 1.41}$ (3) | $4.12_{\pm 2.39}$ (2) | $1.41_{\pm 0.73}$ (2) | **2.17** (1) |

## E  DATASETS, HYPERPARAMETERS, AND PRE-TRAINED MODELS

**Forgetting metric.** Forgetting is defined as $F_\kappa = (1/(\kappa-1)) \sum_{\tau=1}^{\kappa-1} \max_{\tau' \in \{1,\ldots,\kappa-1\}} (\mathrm{acc}(\tau, \tau') - \mathrm{acc}(\tau, \kappa))$ where $\mathrm{acc}(\tau, \kappa)$ denotes model's prediction accuracy on task $\tau$ after learning task $\kappa$. .

**Datasets.** The details of the datasets used in our experiments are summarized in Table 7. The ImageNet-R, ImageNet-A and VTAB5T-Small datasets are taken from `https://github.com/zhoudw-zdw/RevisitingCIL`. CIFAR-100 is accessed directly through torchvision. For VTAB5T-large, we sample 5 datasets from the Visual Task Adaptation Benchmark dataset (Zhai et al., 2019), including EuroSAT, Oxford IIIT Pets, PatchCamelyon, 102 Category Flower Dataset and Resisc45. We build our VTAB-Sim50 from the VTAB5T-large dataset, following the instructions in (Kim et al., 2023). For each task, we retain 50% of its classes and sample the remaining classes from other tasks. As a result, we exclude the PatchCamelyon dataset from VTAB-Sim50, as its two-class structure does not allow for a meaningful sampled dataset.

**Pre-Trained Backbone and Configurations.** For all experiments, we use the ViT-B/16 model (Dosovitskiy, 2020) that was pre-trained on ImageNet-21k and fine-tuned on ImageNet-1K. The rank of all LoRA units in PROTEUS is set to $r = 4$ for the *Uniformly Mild* and the *Varying* settings, and $r = 16$ for the *Uniformly Abrupt* setting. The elastic loss regularization strength $\lambda$ in Eq. 11 is initialized at 0.006 and is progressively reduced by 20% after each task.

Table 7: Dataset Statistics

| CL Datasets | # train samples | # val samples | # classes |
|---|---|---|---|
| CIFAR100 | 50000 | 10000 | 100 |
| Imagenet-R | 24000 | 6000 | 200 |
| Imagenet-A | 5981 | 1519 | 200 |
| VTAB5T-small | 1796 | 8619 | 50 |
| VTAB5T-large | 321406 | 47585 | 196 |
| VTAB-Sim50 | 47328 | 11856 | 144 |

For the regularization weight $\alpha$ in Eq. 11, we conduct grid search with $4$ values from $0$ to $1$ with increment $0.1$ and set it to be $0.8$ which leads to the best performance on our validation data. The user-specified noise level $\gamma$ in Eq. 12 uses the same optimization strategy as in RanPAC. The learning rate varies across datasets, with $1e^{-3}$ for CIFAR-100 and ImageNet-R, $5e^{-4}$ for ImageNet-A and VTAB5T-small, and a smaller $1e^{-4}$ for VTAB5T-large and VTAB-Sim50. The batch size $b$ is adapted accordingly, with larger values (32) for CIFAR-100 and ImageNet-R, ImageNet-A and VTAB5T-small use 16. VTAB5T-large/VTAB-Sim50 use 24. All experiments are run on 2 NVIDIA A40 GPUs with $48$GB (each) in memory.

## F  ADDITIONAL ABLATION STUDIES

**Additional experiments with longer task sequences.** For longer task sequences, we further conducted experiments on 19-task VTAB, 50-task CIFAR-100 and 100-task ImageNet-R. Our results in Table 8 show that PROTEUS consistently outperforms the baselines, demonstrating strong scalability as the task sequence becomes significantly longer. In addition, PROTEUS achieves **91.35%**, **98.15%** and **80.82%** retrieval accuracy on VTAB, CIFAR100 and ImageNet-R respectively, illustrating that its parameter-free retrieval mechanism remains stable and resistant to forgetting even in long-horizon continual fine-tuning.

Table 8: Performance comparison between PROTEUS and parameter-adaptation works on longer task sequences: VTAB (19 tasks), CIFAR-100 (50 tasks), and ImageNet-R (100 tasks).

| Methods | VTAB (19 tasks) | CIFAR-100 (50 tasks) | ImageNet-R (100 tasks) |
|---|---|---|---|
| InfLoRA | 84.15 | 61.93 | 40.6 |
| SD-LoRA | Not ready | 68.19 | 55.08 |
| RanPAC | 93.77 | 89.23 | 72.10 |
| PROTEUS | **94.60** | **91.31** | **75.69** |

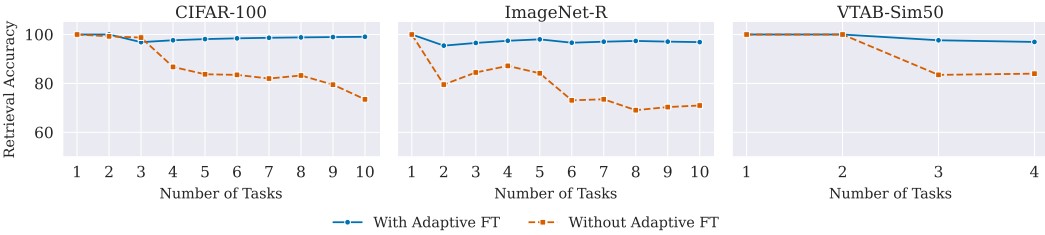

Figure 4: Task retrieval accuracies achieved by PROTEUS and its variant w/o adaptive fine-tuning and enforcing orthogonality among fine-tuning (FT) directions. It can be observed that w/o such conditioning, the task retrieval severely decreases.

To show the empirical robustness of our theoretical bound on achieving high retrieval accuracy with high confidence (see Theorem 3.5) in long sequences of tasks, we setup a 50-task sequence using the CIFAR-100 dataset and show that empirically in Table 9, the cluster separation delta (see Definition

3.3) remains significantly larger than the theoretical minimum to ensure low retrieval error rate (¡ 2%):

Table 9: Report on our empirical cluster separation factor $\delta$ clearly shows that they are greater than minimum requirement to guarantee low retrieval error rate $\epsilon$ across tasks in the Split CIFAR-100 50-task benchmark.

| | Ta. 5 | Ta. 10 | Ta. 15 | Ta. 20 | Ta. 25 | Ta. 30 | Ta. 35 | Ta. 40 | Ta. 45 | Ta. 50 |
|---|---|---|---|---|---|---|---|---|---|---|
| Empirical $\delta$ | 163.01 | 105.02 | 163.31 | 159.36 | 140.94 | 98.93 | 128.74 | 88.84 | 145.01 | 143.41 |
| Theoretical $\delta$ | 5.17 | 6.60 | 5.67 | 5.88 | 5.71 | 6.24 | 4.68 | 6.31 | 5.40 | 6.59 |

**Additional Results on the Orthogonality between Old and New LoRA Units.** This section provides additional empirical verification that the orthogonality constraint in our adaptive fine-tuning loss in Eq. 11 is indeed satisfied in the numerically optimized solution. In particular, we visualize the cosine distance between the corresponding LoRA updates across all tasks in VTAB5T-Large (Fig. 6) and ImageNet-A (Fig. 7). In both experiments, we observe that the cosine distances between the corresponding tuning directions $\Delta\omega_\tau^\perp$ of different tasks are vanishingly small, confirming their orthogonality. This observation also remains consistent across various experiments where the adaptive LoRA fine-tuning is applied at different attention layers. The enforced orthogonality in turn helps improve the task retrieval as previously demonstrated in Section 3.

**Understanding Sparsity in the Knowledge Transfer Matrix.** The norm regularization term in Eq. 11 enforces sparsity, encouraging $S$ to focus on the most relevant tasks and avoid excessive reliance on past ones. Fig. 5(a-b) provides heatmap visualizations for the magnitudes of non-zero (diagonal) entries on $S$ on ImageNet-R at task no. 3 and 9, showing the selective relevance of past LoRA units to the respective current tasks. Each heatmap has 12 rows, indexing separate LoRA pools for different attention blocks. The no. of columns indicate the no. of LoRA units per pool. Note that PROTEUS adds 4 units per task, which increases the no. of columns when $\tau$ increases. In all heatmaps, we observe that $S$ exhibits a highly sparse structure as expected.

We further conduct an experiment to analyze the impact of this regularizer. Specifically, we compare the standard setting of PROTEUS against a variant with $\lambda = 0$, which effectively removes the norm penalty on $S$. Fig. 5(c) recreates the heatmap visualization of non-zero terms in $S$ at the final task ($\tau = 9$), illustrating a much denser usage of previous LoRA units compared to the version with sparsity enforced via Eq. 11 (shown in Fig. 5). The corresponding performance is shown in the second last row of Table 10, severely degraded to $30.49\%$ after training. This degradation can be attributed to the excessive and unfiltered reuse of prior LoRA units, including those that negatively affect performance, making the model harder to optimize and reducing its effectiveness.

**Comparing to Top-K Gaussian signature function.** Since selecting the closest Gaussian component yields strong performance, we perform an ablation study on the use of Top-K Gaussians instead. Given a new input $x$, we compute the likelihood of $h(x)$ under all Gaussian components for each task and aggregate the Top-K likelihoods by summing them, producing a task-specific signal. The task with the largest signal is then inferred as the task identity. We show the results for $K = 1, 3, 5$ on two VTAB5T datasets in Table 11, which confirms that the accuracy at $K = 1$ is the best.

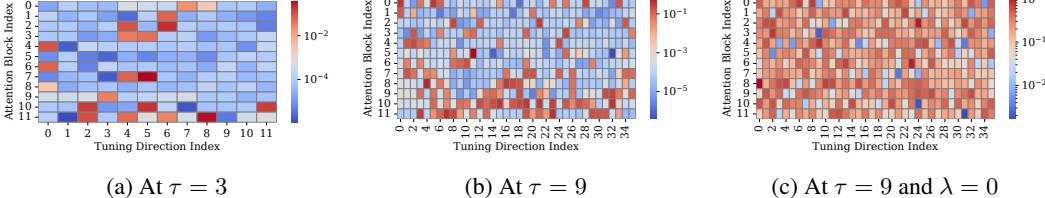

(a) At $\tau = 3$       (b) At $\tau = 9$       (c) At $\tau = 9$ and $\lambda = 0$

Figure 5: Heatmap visualizations of $S$ at different iterations on ImageNet-R with $\lambda > 0$ and $\lambda = 0$.

**Advantage of DP-GMM over K-Means.** Conceptually, DP-GMM offers two key advantages over K-means clustering. First, it automatically adapts the number of clusters, which is crucial in contin-

Table 10: PROTEUS's classification and retrieval accuracy achieved with various settings of $S$ and $\lambda$ on ImageNet-R dataset.

| Configs | Classification | Retrieval |
|---------|---------------|-----------|
| $S = 0$ | 78.98 | 87.05 |
| $S = I$ | 30.38 | 41.45 |
| $S_*, \lambda = 0$ | 30.49 | 41.97 |
| $S_*, \lambda > 0$ | 82.17 | 96.03 |

Table 11: PROTEUS's classification accuracy achieved with Top-K Gaussians retrieval where $K = 1, 3, 5$ on two VTAB datasets.

| K | VTAB5T-Small | VTAB5T-Large |
|---|-------------|-------------|
| 1 | **94.76** | **89.32** |
| 3 | 93.72 | 76.50 |
| 5 | 92.34 | 76.25 |

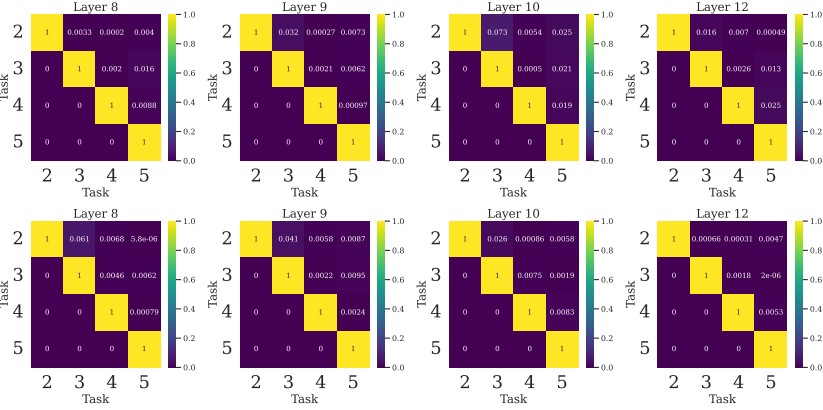

Figure 6: Pairwise cosine similarities of LoRA fine-tuning directions $\Delta\omega_\tau^\perp$ across tasks $\tau$ of VTAB5T-Large for the query (top) and value (bottom) matrices of the multi-head self-attention (MHSA) at various layers. For clearer visualization, we display only the upper half of the cosine similarity matrices since they are symmetric.

ual learning where tasks vary widely in complexity and semantics. Second, DP-GMM's probabilistic modeling allows PROTEUS to compute principled likelihood scores for test inputs, and leverage those for retrieval. To empirically demonstrate the importance of the DP-GMM module, we provide an additional experiment in which DP-GMM is replaced with K-Means (using $K = 20$) in our pipeline. This simple change resulted in up to **13/18% drop** in classification/retrieval accuracy (see the Table 12 below).

Table 12: Performane Comparison of DP-GMM and K-Means on ImageNet-R and ImageNet-A

| Datasets | DP-GMM | | K-Means | |
|----------|--------|------|---------|------|
| | Classifier | Retrieval | Classifier | Retrieval |
| ImageNet-R | **82.69** | **96.89** | 81.15 | 92.75 |
| ImageNet-A | **63.77** | **77.55** | 55.34 | 63.63 |

# G  GPU FOOTPRINT DURING TRAINING

Fig. 8 compares PROTEUS's GPU memory overhead during training to those of other baseline methods on ImageNet-A. Since PEFT-based CL methods expand their memory to accomodate increasing tasks, we report the maximum memory incurred during training of each method. The results show that PROTEUS requires less GPU memory than the competing approaches while delivering superior performance, as highlighted in Table 1. Notably, PROTEUS's memory consumption is significantly lower than RanPAC's while achieving better accuracy than RanPAC.

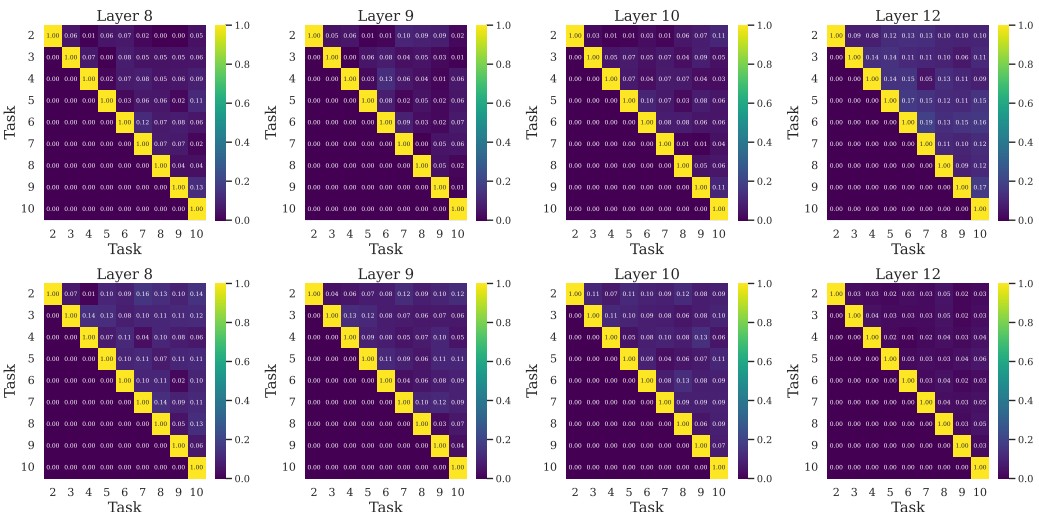

Figure 7: Pairwise cosine similarities of LoRA fine-tuning directions $\Delta\boldsymbol{\omega}_\tau^\perp$ across tasks $\tau$ of ImageNet-A for the query (top) and value (bottom) matrices of the multi-head self-attention (MHSA) at various layers. For clearer visualization, we display only the upper half of the cosine similarity matrices since they are symmetric.

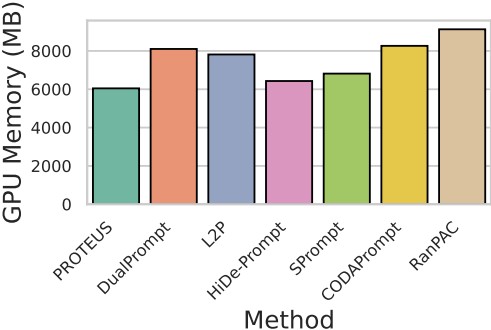

Figure 8: GPU consumption (MB) of various baselines. PROTEUS consumes least GPU.

## H DETAILED ANALYSIS ON MEMORY COST.

To further provide analysis about total memory cost induced during training process, we first provide an example on common settings. Choosing a low LoRA rank $r = 4$ per task, i.e., 4 rank-1 LoRA units. As the embedding dimension of each transformer block in ViT is 768 and there are 12 such blocks, each rank-1 LoRA unit comprises 24 vectors of size 768 x 1. In addition, the sizes of the mean vector and covariance matrix of each resulting input embedding cluster under each rank-1 LoRA unit are 768 x 1 and 768 x 768 respectively. As our non-parametric clustering algorithm is configured with at most 20 such clusters, each task incurs at most 24 * 4 * 768 * 4 (bytes) = 288KB and $(768 + 768 * 768) * 4 * 20 < 46$MB to store 4 rank-1 LoRA units and the (mean, covariance) signature for each resulting input embedding cluster, respectively. This amounts to 48MB of overhead per task which is quite affordable given the capacity of modern computers. Hence, a large sequence of 100 tasks will incur no more than 4.8GB of overhead.

Alternatively, we can also enhance the current approach via using a schedule of exponentially decaying LoRA rank. Here, we assume that with an increasing number of observed tasks, the incremental information provided by each newly arriving task diminishes. Hence, the LoRA rank for latter tasks can be reduced accordingly. This ensures the total parameter cost remains strictly bounded. In particular, let $r_m$ denote the LoRA rank at task $m$, we choose an exponential decay strategy such as: $r_m = r_o \exp\left(-\alpha(m-1)\right)$ so the sequence $r_1, \ldots, r_m$ forms a geometric series or $\sum_{m=1}^{\infty} r_m = \frac{r_0}{1-\exp{-\alpha}} < \infty$. Thus, even as $M \to \infty$, the total LoRA memory remains finite.

Let $\tau$ be the number of clusters of each task , we introduce $\tau * M$ clusters after observing M tasks. Let c be the memory cost per rank-1 LoRA unit, and v be the memory cost per cluster. Following the above calculations:

- c = 24 * 768 * 4 * 4 (bytes) = 288KB = 0.288MB

- $v = (768 + 768 * 768) * 4 (bytes) < 2.4$ MB

The total memory cost is then upper-bounded via: $\sum_{m=1}^{\infty} c * r_m + v * \tau * M \leq H$.

Choosing $r_0 = 10, \alpha = 0.05, \tau = 20, M = 20$, we have H = 1019.05MB < 1GB. Using this budget-constrained configuration on a 20-task simulation of the CIFAR100 dataset, we obtained promising results as reported below. Despite the tight budget, the performance of the budget-constrained PROTEUS matches that of its unconstrained version (fixed r = 10) as Table 13:

Table 13: Performance comparison on CIFAR100 20-task setting between PROTEUS under budget-constrained strategy and original unconstrained version.

| Methods | Ta. 2 | Ta. 4 | Ta. 6 | Ta. 8 | Ta. 10 | Ta. 12 | Ta. 14 | Ta. 16 | Ta. 18 | Ta. 20 |
|---|---|---|---|---|---|---|---|---|---|---|
| Budget-Constrained | 99.1 | 98.6 | 97.9 | 97.57 | 96.82 | 96.46 | 96.22 | 96.1 | 96.21 | 95.43 |
| Unconstrained | 99.1 | 98.4 | 97.7 | 97.17 | 96.74 | 96.50 | 96.54 | 96.51 | 96.60 | 95.65 |

In a more challenging setting on ImageNet-R with $r_0 = 10, \alpha = 0.1, \tau = 50$ which enforces a maximum memory budget H < 2.5GB, the budget-constrained PROTEUS still manages to retain strong performance compared to its unconstrained version (fixed r = 10) as reported in Table 14 below.

Table 14: Performance comparison on ImageNet-R 20-task setting between PROTEUS under budget-constrained strategy and original unconstrained version.

| Methods | Ta. 2 | Ta. 4 | Ta. 6 | Ta. 8 | Ta. 10 | Ta. 12 | Ta. 14 | Ta. 16 | Ta. 18 | Ta. 20 |
|---|---|---|---|---|---|---|---|---|---|---|
| Budget-Constrained | 87.55 | 86.27 | 82.43 | 82.57 | 82.85 | 82.86 | 81.31 | 79.53 | 80.04 | 79.13 |
| Unconstrained | 90.28 | 90.03 | 89.36 | 88.39 | 87.53 | 86.92 | 87.20 | 86.46 | 86.74 | 86.65 |

These results consistently demonstrate that our exponential-decay LoRA strategy can rigorously bound memory growth while preserving the optimal performance of the unconstrained version (with mild degradation). This confirms that PROTEUS remains practical, affordable (in memory cost), and scalable in long-term continual learning scenarios.

# I  RELATED WORKS

**Continual Learning.** CL aims to learn new tasks effectively without sacrificing the knowledge learned in previous tasks. Conventional CL methods fall into 3 main categories: rehearsal-based, regularization-based and architecture-based. Rehearsal-based CL methods (Rolnick et al., 2019; Shin et al., 2017; Bang et al., 2021; Buzzega et al., 2020; Aljundi et al., 2019b;a; Hayes et al., 2019; Hou et al., 2019; Lopez-Paz & Ranzato, 2017; Von Oswald et al., 2019; Van de Ven et al., 2020) utilize memory buffers to store past examples of previous tasks in order to reduce forgetting, while regularization based methods (Titsias et al., 2019; Pan et al., 2020; Chaudhry et al., 2018; Kirkpatrick et al., 2017) add regularization terms to constrain the weight updates such that they do not interfere significantly with the knowledge learned from previous tasks.

On the other hand, architecture-based methods (Mallya et al., 2018; Ebrahimi et al., 2020; Lee et al., 2020) introduce new parameters for each new task, allowing the model to specialize for new tasks without disrupting the learned representations of previous ones. However, traditional methods for CL are not suitable for adapting large pretrained models, which is the focus of this paper, due to their need to fully fine-tune the pre-trained models.

**Parameter Efficient Continual Learning.** With the advent of foundation models, various parameter efficient finetuning methods have been applied to efficiently adapt foundation models to new unseen task with a small set of learnable parameters in CL settings. Previous works on prompt based CL methods have focused on how to further design and retrieve prompts to reduce catastrophic forgetting. Particularly, L2P (Wang et al., 2022c), DualPrompt (Wang et al., 2022b) and CODAPrompt (Smith et al., 2023) leverage prompt tuning for ViT in CL setting. Based on these works, HiDe-Prompt (Wang et al., 2023a) enhances the performance by storing sample features alongside with learning the prompts.

On the other hand, LoRA-based methods such as InfLoRA (Liang & Li, 2024b), Online-LoRA (Wei et al., 2024), SD-LoRA (Wu et al., 2025b) leverage LoRA units to adapt to new tasks in CL. InfLoRA and SD-LoRA apply the final accumulated LoRA weight to all tasks, hence they do not have any retrieval components, comprimising task-specific performance. More recently, O-LoRA (Wang et al., 2023b) also proposes a different approach to continually learn new LoRA units which are orthogonal to previous units for each new task. However, unlike our approach, O-LoRA uses all LoRA units during inference rather than carefully retrieving the most relevant units for each new input. This leads to a lack of representation adaptability which is similar to RanPAC (McDonnell et al., 2023) and consequently results in worse performance than our method – see Table 16.

Moreover, we note that none of the existing parameter-adaptation methods for continual finetuning adopts retrieval, so adding existing parametric retrieval on the other hand will not work out of straightforward integration. To demonstrate this, we conducted an additional experiment on ImageNet-R, combining RANPAC McDonnell et al. (2023) (which is best among state-of-the-art parameter-adaptation methods) with the parametric retrieval mechanism of HiDE (Wang et al., 2023a) to retrieve the most relevant LoRA unit for each input during test time. Our new results in Table 15 show that this vanilla integration of parametric retrieval expectedly degrades the performance of RANPAC due to forgetting (as evidenced by weak retrieval accuracy). On the other hand, PROTEUS with guaranteed parameter-free retrieval (see Theorem 3.5 and Fig. 2) achieves 96% of retrieval accuracy and consequently, significantly better performance (see table below).

Table 15: Performance comparison between PROTEUS and two versions of RanPAC: the original paper (RanPAC (wo/ Retrieval)) and its variant using the parametric retrieval mechanism of HiDE (Wang et al., 2023a) to retrieve the most relevant LoRA unit for each input during test time (RanPAC (w/ Retrieval))

| Method | Performance | Retrieval |
|---|---|---|
| RanPAC (w/ Retrieval) | 71.79 | 75.18 |
| RanPAC (wo/ Retrieval) | 78.08 | N/A |
| PROTEUS | **82.17** | **96.02** |

Alternatively, another recent method, MELO (Yu et al., 2024), focuses instead on a different scheme of parameter-free retrieval approach based on clustering. However, it relies on carefully tuned clustering hyperparameters which fluctuates wildly across different pre-trained backbones. Furthermore, MELO was developed specifically for NLP applications using T5/BERT/GPT2 backbone, leaving it uncertain how to adapt its hyperparameters for vision tasks with ViT backbone. It also lacks mechanisms to mitigate knowledge interference (e.g., orthogonality constraints). In contrast, PROTEUS mitigates knowledge interference via both adaptive knowledge selection and orthogonal constraints, which results in significant performance improvement over both O-LoRA and MELO – see Table 16.

We evaluate the performance of O-LoRA and MELO on vision tasks by adapting their designs to the PROTEUS framework for a fair comparison. For O-LoRA, we remove the retrieval component and disable knowledge reuse by setting $S = 0$ while replacing our regularization with O-LoRA's. At inference time, all LoRA units are used rather than being retrieved selectively for each input. In contrast, for MELO, we replace its retrieval mechanism with our own and remove both the orthogonality constraints and the knowledge reuse component during training (setting $S = 0$), thus reproducing MELO's behavior for a fair evaluation. The reported results in Table 16 show that PROTEUS achieves much better performance than both O-LoRA and MELO across multiple benchmarks.

Table 16: Performance comparison between our proposed method PROTEUS and its variants incorporating and adapting the continual fine-tuning approaches in O-LoRA (Wang et al., 2023b) and MELO (Yu et al., 2024). We note that these approaches were developed for NLP settings with T5/BERT/GPT2 backbone and need to be adapted to our CV settings with the ViT backbone.

| CL Datasets | PROTEUS | O-LoRA | MELO |
|---|---|---|---|
| CIFAR-100 | **94.10** | 54.55 | 92.98 |
| ImageNet-R | **82.17** | 53.93 | 78.98 |
| ImageNet-A | **62.76** | 32.11 | 38.47 |
| VTAB5T-small | **92.34** | 87.69 | 87.67 |
| VTAB-Sim50 | **95.89** | 90.64 | 91.07 |

**Parameter Efficient Fine-tuning (PEFT).** Full fine-tuning of modern large language models incurs a substantial amount of computational and storage resources and runs the risk of overfitting. To mitigate these, PEFT methods have been proposed, which attain or even outperform the performance of traditional fine-tuning methods while learning only a small number of parameters. Notably, prompt-tuning (Qin & Eisner, 2021; Liu et al., 2021) and prefix-tuning (Li & Liang, 2021) prepend learnable parameters to the input of the transformer.

Likewise, adapters introduces learnable modules to the layers of transformers for each new task. On the other hand, low-rank adaptation (LoRA) (Hu et al., 2021) decomposes the weight update matrix into lower-ranked matrices and fine-tunes these low rank components. PEFT methods have also been proposed for vision tasks, such as visual prompt tuning (Jia et al., 2022), which achieves comparable or better performance than full fine-tuning. However, PEFT methods are usually applied in static data settings while real-world scenarios require models to continuously adapt to new changes, necessitating approaches that allows PEFT to be applied dynamically over time.

## J    TASK RETRIEVAL OVERHEAD

In our largest experiment on ImageNet-R with the maximum number of Gaussian components $K = 100$, the average inference cost is on average 2.8 seconds per batch (with batch size = 32). To quantify the trade-off between predictive performance and inference efficiency, we introduce the metric *Retrieval Accuracy Gain* (%/sec):

$$\text{Retrieval Accuracy Gain} = \frac{\text{Retrieval Accuracy}_{\text{PROTEUS}} - \text{Retrieval Accuracy}_{\text{Baselines}}}{\text{Inference Time}_{\text{PROTEUS}} - \text{Inference Time}_{\text{Baselines}}}. \quad (33)$$

This metric quantifies the retrieval accuracy improvement per second of inference time, highlighting the trade-off between retrieval effectiveness and computational cost. A higher value indicates that PROTEUS achieves greater retrieval accuracy gains for each second spent during inference. Table 17 presents the Retrieval Accuracy Gain of PROTEUS compared to other retrieval-based baselines on ImageNet-R. Notably, PROTEUS achieves the highest improvement, with a gain of 22.29% retrieval accuracy per second over Dual-Prompt.

This result demonstrates the efficiency of PROTEUS's retrieval mechanism, offering a compelling balance between performance and inference overhead in retrieval-based continual learning. By leveraging task-specific representations and parameter-free retrieval, PROTEUS achieves significant gains in accuracy due to a thorough mitigation of forgetting during test time, as demonstrated in our experiments. It is also worth noting that the retrieval process across different tasks can be parallelized, which helps reduce the inference-time overhead in practice. We believe that further optimizing this trade-off presents an exciting direction for future work.

## K    PARAMETER COMPARISON

Table 18 reports the total number of parameters introduced during training on VTAB-Sim50 by each baseline, using the optimal configuration provided in their released code.

Table 17: Retrieval Accuracy Gain of PROTEUS over retrieval-based methods on ImageNet-R.

| CL Methods | Dual-Prompt | SPrompt | HiDE |
|---|---|---|---|
| **Retrieval Accuracy Gain (% / sec)** | 22.29 | 20.78 | 19.07 |

Table 18: No. of parameters in the optimal configuration of each baseline methods (reported in their papers and/or released code) and their corresponding performance on the VTAB-Sim50 dataset.

| CL Methods | #Parameters | Performance |
|---|---|---|
| L2P | $156,868$ | $80.24_{\pm1.46}$ |
| Dual-Prompt | $602,308$ | $87.71_{\pm0.33}$ |
| SPrompt | $1,650,628$ | $90.69_{\pm0.22}$ |
| CoDAPrompt | $3,950,786$ | $86.01_{\pm1.91}$ |
| AdaPromptCL | $18,773,186$ | $77.86_{\pm1.62}$ |
| RanPAC | $213,505$ | $90.22_{\pm0.24}$ |
| InfLoRA | $251,992$ | $91.25_{\pm0.25}$ |
| SD-LoRA | $464,786$ | $84.91_{\pm1.95}$ |

The number of parameters in the knowledge transfer matrix $S$ of PROTEUS is given by no. LoRA-attention matrices $\times$ no. blocks applied $\times$ no. current LoRA units. In our setup, LoRA is applied to query and value matrices across all 12 blocks. The no. of current LoRA units is given by $r \times$ no. tasks seen, where $r$ is the LoRA rank.

Overall, the number of parameters in $S$ remains relatively small throughout training. For example, in the optimal configuration on VTAB-Sim50, we achieve superior performance over all baselines with just **258,192** parameters, making PROTEUS more compact than most previous methods. On the same task, a lightweight variant of PROTEUS, which uses LoRA on the first 8 Transformer blocks with only **209,092** parameters, achieves **95.44**% accuracy, surpassing all baselines while using fewer parameters than most (except L2P). To enable a direct comparison with L2P, which has the fewest parameters among baselines, we further reduce the number of parameters of PROTEUS to **159,940** by applying LoRA to just the first 4 Transformer blocks. Even under this constraint, PROTEUS achieves **88.63**% accuracy, outperforming L2P at equivalent parameter complexity.

## L  ABLATION ON REGULARIZATION PARAMETER $\alpha$ IN EQ. 11

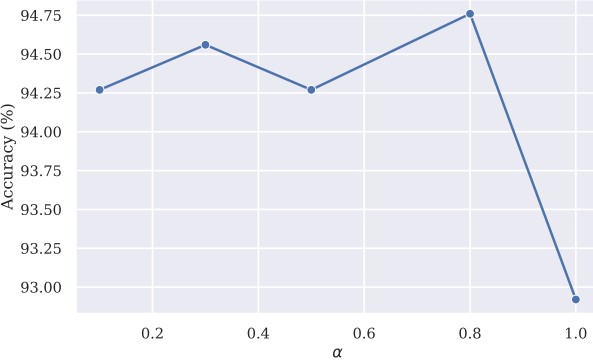

Figure 9: Performance of PROTEUS as $\alpha$ varies from 0.1 to 1.0.

In this section, we investigate the sensitivity of PROTEUS's performance to the regularization weight $\alpha$. As defined in Eq.equation 11, $\alpha$ controls the trade-off between the $\ell_1$ and $\ell_2$ norm penalties where larger values of $\alpha$ emphasize sparsity while smaller values encourage stability and smoothness in the selection matrix $S$. To ablate the impact of this trade-off, we fix all other hyperparameters as

specified in Appendix E and vary $\alpha$ across the set $\{0.1, 0.3, 0.5, 0.8, 1.0\}$. We conduct this ablation study on the VTAB5T-small benchmark.

As shown in Fig. 9, the best performance is achieved at $\alpha = 0.8$, reaching a peak accuracy of **94.76**%. In contrast, setting $\alpha = 1.0$, which corresponds to using only the $\ell_1$ norm, leads to a drop in accuracy. This underscores the importance of incorporating the $\ell_2$ term to preserve stability and smoothness in $S$. The results demonstrate that a balanced combination of sparsity and stability $(0 < \alpha < 1)$, such as elastic loss, leading to more effective and generalizable representations.

## M    LIMITATIONS

Despite the performance advantage of PROTEUS in class-incremental settings, we foresee that the retrieval-based task signature construction will face some challenges when task boundaries are ambiguous or data distributions overlap in more complex settings such as task-free or online continual learning. Extending our framework to these settings will be part of our future work.

