# OpenReview forum: "Continual Fine-Tuning with Provably Accurate and Parameter-Free Task Retrieval"
_ICLR.cc/2026/Conference — Submitted to ICLR 2026_

### Official Review · Reviewer_JdHy · 2025-10-29

**Soundness:** 3
**Presentation:** 3
**Contribution:** 3
**Rating:** 6
**Confidence:** 4

**Summary:**

The paper introduces PROTEUS, a continual fine-tuning framework that achieves provably accurate, parameter-free task retrieval. It bridges the gap between input-adaptation and parameter-adaptation methods by combining adaptability in test-time representation with retrieval-free prediction. The method derives theoretical bounds linking retrieval accuracy to the clustering structure of task-specific representations, ensuring low retrieval error through well-separated clusters. PROTEUS incorporates an adaptive fine-tuning mechanism that promotes orthogonal and informative updates across tasks, preserving prior knowledge while improving discrimination among task embeddings.

**Strengths:**

1. The paper introduces a parameter-adaptation method that uniquely combines adaptive use of input embeddings with parameter-free task retrieval, bridging the gap between input-adaptation and parameter-adaptation paradigms.

2. The authors provide a theoretical analysis that formally connects retrieval accuracy to structural properties of task-specific representation clusters. This foundation is concretely realized through the adaptive module composition strategy, which enforces orthogonality and selective knowledge transfer, ensuring both preservation and complementarity of prior knowledge.

3. The clustering-based retrieval mechanism offers a clear, interpretable, and scalable solution for adaptive test-time inference, improving retrieval stability and reducing forgetting.

**Weaknesses:**

1. As noted, the method generates a distinct LoRA update for each task, which can become memory-intensive when scaling to long sequences or high-dimensional backbones. Although the authors argue that LoRA units are lightweight, the cumulative storage of adaptation modules and their associated signature distributions may still pose practical challenges.

2. The Gaussian clustering assumption is a somewhat strong theoretical simplification in the paper. In reality, feature distributions might be non-Gaussian or overlap between tasks, violating the conditions under which PROTEUS can provably bound the retrieval error.

**Questions:**

1. The paper presents results primarily on 10-task sequences (e.g., Split CIFAR-100, ImageNet-R, and VTAB benchmarks), but continual fine-tuning is especially challenging when the number of tasks grows larger. It would be helpful if the authors could provide or discuss results for shorter (e.g., 5) and longer (e.g., 20 or 50) task sequences, to better understand how PROTEUS scales in performance, retrieval accuracy, and memory usage as the task sequence length increases.

2. Could the authors consider introducing a fixed memory budget and examining how performance degrades under such constraints?

---

> ### Author Response · Authors · 2025-11-21
> **Official Comment**
>
> We thank the reviewer for the positive review and acknowledging our contribution, presentation and soundness.
>
> **Memory Cost**
>
> We appreciate the reviewer’s comment on the growing memory cost. This is indeed a broad issue shared by most CFT methods which is however unavoidable as it is impractical to expect fixed memory footprint while allowing the no. tasks to grow indefinitely.
>
> In our previous experiment, we minimize this impact via choosing a low LoRA rank r = 4 per task, i.e., 4 rank-1 LoRA units. As the embedding dimension of each transformer block in ViT is 768 and there are 12 such blocks, each rank-1 LoRA unit comprises 24 vectors of size 768 x 1. In addition, the sizes of the mean vector and covariance matrix of each resulting input embedding cluster under each rank-1 LoRA unit are 768 x 1 and 768 x 768 respectively. As our non-parametric clustering algorithm is configured with at most 20 such clusters, each task incurs at most 24 * 4 * 768 * 4 (bytes) = 288KB and (768 + 768 * 768) * 4 * 20 < 46MB to store 4 rank-1 LoRA units and the (mean, covariance) signature for each resulting input embedding cluster, respectively. This amounts to 48MB of overhead per task which is quite affordable given the capacity of modern computers. For example, a large sequence of 100 tasks will incur no more than 4.8GB of overhead while even a low-end laptop now has at least 8GB RAM.
>
> **Fixed Memory Budget**
>
> Alternatively, we can also enhance the current approach via using a schedule of exponentially decaying LoRA rank. Here, we assume that with an increasing number of observed tasks, the incremental information provided by each newly arriving task diminishes. Hence, the LoRA rank for latter tasks can be reduced accordingly. This ensures the total parameter cost remains strictly bounded. In particular, let r_m denote the LoRA rank at task m, we choose an exponential decay strategy such as: $r_m = r_0 e^{-\alpha (m-1)}.$, so the sequence $r_1, \dots, r_m$ forms a geometric series or
>
> $$
> \sum_{m=1}^{\infty} r_m = \frac{r_0}{1 - e^{-\alpha}} < \infty.
> $$
>
> Thus, even as $M\to \infty$, the total LoRA memory remains finite.
>
> Let $\tau$ be the number of clusters of each task , we introduce $\tau * M$ clusters after observing $M$ tasks.\
>
> Let $c$ be the memory cost per rank-1 LoRA unit, and $v$ be the memory cost per cluster. Following the above calculations:
> $c = 24 * 768 * 4 * 4 (bytes) = 288KB = 0.288MB$
>
> $v = (768 + 768 * 768 ) * 4 (bytes) < 2.4 MB.$
>
> The total memory cost is then upper-bounded via:
>
> $\left( \sum_{m=1}^{\infty} c * r_m \right) + v * \tau * M \le H.$
>
> Choosing $r_0 = 10,  \alpha= 0.05, \tau = 20, M = 20$, we have $H = 1019.05MB < 1GB$. Using this budget-constrained configuration on a 20-task simulation of the CIFAR100 dataset, we obtained promising results as reported below. Despite the tight budget, the performance of the budget-constrained PROTEUS matches that of its unconstrained version (fixed $r = 10$):
>
> |Methods|Task 2|Task 4|Task 6|Task 8|Task 10|Task 12|Task 14|Task 16|Task 18|Task 20|
> |------|------|------|-------|-------|-------|-------|-------|-------|-------|-------|
> |Budget-Constrained|99.1|98.6|97.9|97.57|96.82|96.46|96.22|96.1|96.21|95.43|
> |Unconstrained|99.1|98.4|97.7| 97.17|96.74|96.50|96.54|96.51|96.60|95.65|
>
> In a more challenging setting on ImageNet-R with $r_0 = 10,  \alpha= 0.1, \tau = 50$ which enforces a maximum memory budget $H < 2.5GB$, the budget-constrained PROTEUS still manages to retain strong performance compared to its unconstrained version (fixed $r = 10$) as reported below.
>
> |Methods|Task 2|Task 4|Task 6|Task 8|Task 10|Task 12|Task 14|Task 16|Task 18|Task 20|
> |------|------|------|-------|-------|-------|-------|-------|-------|-------|-------|
> |Budget-Constrained|87.55|86.27|82.43|82.57|82.85|82.86|81.31|79.53|80.04|79.13|
> |Unconstrained|90.28|90.03|89.36|88.39|87.53|86.92|87.20|86.46|86.74|86.65|
>
> These results consistently demonstrate that our exponential-decay LoRA strategy can rigorously bound memory growth while preserving the optimal performance of the unconstrained version (with mild degradation). This confirms that PROTEUS remains practical, affordable (in memory cost), and scalable in long-term continual learning scenarios.

---

> > ### Author Response · Authors · 2025-11-21
> > **Official Comment**
> >
> > **Additional experiments with shorter and longer task sequences**
> >
> > For shorter task sequences, we have already evaluated PROTEUS on **small VTAB variants** with 4-task and 5-task settings (e.g., VTAB-Sim50 and VTAB-Small; see Table 1 and Fig. 3). Across these benchmarks, PROTEUS consistently outperforms or matches state-of-the-art baselines, indicating strong performance even with very short task horizons.
> >
> > For longer task sequences, we further conducted experiments on 19-task VTAB, 50-task CIFAR-100 and 100-task ImageNet-R. Our results show that PROTEUS consistently outperforms the baselines, demonstrating strong scalability as the task sequence becomes significantly longer.
> >
> > In addition, PROTEUS achieves **91.35%, 98.15%** and **80.82%** retrieval accuracy on VTAB, CIFAR100 and ImageNet-R respectively, illustrating that its parameter-free retrieval mechanism remains stable and resistant to forgetting even in long-horizon continual fine-tuning.
> >
> > | Methods  | VTAB (19 tasks) | CIFAR-100 (50 tasks) | ImageNet-R (100 tasks) |
> > |----------|------------------|------------------------|--------------------------|
> > | InfLoRA  | 84.15            | 61.93                 | 40.67                   |
> > | SD-LoRA  | Not ready        | 68.19                 | 55.08                   |
> > | RanPAC   | 93.77            | 89.23                 | 72.10                   |
> > | PROTEUS  | **94.60**        | **91.31**             | **75.69**               |
> >
> > **Feature distributions might be non-Gaussian**
> >
> > We acknowledge that real embedding distributions might not be Gaussian. It is however well-established that any distributions can be approximated well with a mixture of Gaussian with a sufficiently large number. of Gaussian. This is enabled via the use of DP-GMM which is a non-parametric clustering method that automatically adapts the number of Gaussians to fit each distribution. This design allows the model to handle complex, non-Gaussian distribution approximately well as verified by the overall superior performance over existing SOTA baselines.
> >
> > Furthermore, our results on the most challenging VTAB benchmark that comprises a wide range of tasks with large semantic gaps (whose input embeddings are likely complex, non-Gaussian) show that PROTEUS consistently performs better than existing SOTA baselines (see Table 1). This attests to the reliability of our parameter-free (or non-parametric) retrieval mechanism even when the real embedding distributions are not Gaussian.
> >
> >  **Overlap between tasks**
> >
> > Regarding potential task overlap, we emphasize that our experiments follow the standard **class-incremental** protocol, where class labels do not repeat across tasks. This is consistent with the common setup in existing SOTA methods such as L2P, DualPrompt, HiDE, RanPAC, InfLoRA, and SDLoRA. Within this setting, PROTEUS achieves the highest retrieval accuracy (Table 3), demonstrating that our clustering mechanism is effective in practice. We acknowledge that overlapping label spaces may introduce ambiguity (Appendix L) but developing CFT methods to handle such scenarios is out of the scope of the current paper. It will be a promising direction for future work. We thank the reviewer for this suggestion.
> >
> > **Retrieval bound remains valid in settings with long sequences of tasks**
> >
> > To show the empirical robustness of our theoretical bound on achieving high retrieval accuracy with high confidence (see Theorem 3.5) in long sequences of tasks, we setup a 50-task sequence using the CIFAR-100 dataset and show that empirically, the cluster separation delta (see Definition 3.3) remains significantly larger than the theoretical minimum to ensure low retrieval error rate (< 2%):
> >
> > ||Task 5|Task 10|Task 15|Task 20|Task 25|Task 30|Task 35|Task 40|Task 45|Task 50|
> > |------|------|------|-------|-------|-------|-------|-------|-------|-------|-------|
> > Empirical delta|163.01|105.02|163.31|159.36|140.94|98.93|128.74|88.84|145.01|143.41|
> > Theoretical delta|5.17|6.60|5.67|5.88|5.71|6.24|4.68|6.31|5.40|6.59|
> >
> > Overall, our empirical results consistently show that PROTEUS remains robust under complex, real-world feature distributions, supporting the validity and practical strength of our design.

---

### Official Review · Reviewer_as1a · 2025-10-30

**Soundness:** 3
**Presentation:** 2
**Contribution:** 3
**Rating:** 6
**Confidence:** 4

**Summary:**

This paper addresses the challenge of catastrophic forgetting in continual fine-tuning (CFT), specifically tackling the limitations of existing input-adaptation and parameter-adaptation methods. The authors argue that input-adaptation methods (e.g., prompt retrieval) suffer from "retriever forgetting" , while parameter-adaptation methods (e.g., RanPAC) lack representation adaptability at test time

**Strengths:**

1. Clearly identifies the problem of retriever forgetting in prompt/parameter-pool based CFT methods and proposes a novel parameter-free retrieval mechanism as a direct solution.

2. Theoretical Foundation: Provides a non-trivial theoretical analysis linking the retrieval error rate to geometric properties (cluster separation factor $\delta$) of the learned representation signatures. This offers valuable insight and principled guidance for the algorithmic design. Theorem 3.4 and 3.5 are significant.

3. Achieves SOTA performance on several challenging CL benchmarks, including those with large semantic gaps like VTAB , significantly outperforming both prompt-based and other LoRA-based CL methods.

**Weaknesses:**

1. High System Complexity: The overall PROTEUS framework is quite intricate, involving adaptive LoRA with orthogonality constraints, non-parametric GMM fitting (DP-GMM) for potentially many components per task, storing these GMM parameters as signatures, computing likelihoods against all signatures during retrieval, and finally performing LDA prediction. This complexity raises concerns about implementation difficulty, computational overhead (especially GMM fitting and retrieval), and potential fragility.

2. Insufficient Ablation of Design Choices: While key components (retrieval, adaptive FT) are ablated, the paper could benefit from more fine-grained ablations. For example, why is DP-GMM necessary? Would simpler clustering (e.g., k-means for centroids) or simpler signatures (e.g., just means) suffice? Is the combination of $l_1/l_2$ regularization 95and orthogonality 96 both necessary in the LoRA update? Justifying the necessity of each complex piece versus simpler alternatives would strengthen the paper.

3. Reliance on Distributional Assumptions: The method relies on the assumption that task-specific embeddings $h_k(x)$ can be well-modeled by GMMs. The theoretical bounds (Assumption A.1) also depend on this. While GMMs are flexible, the quality of fit and the resulting cluster separation might degrade if the true embedding distributions are highly non-Gaussian or have complex structures, potentially impacting retrieval accuracy and invalidating the theoretical guarantees

**Questions:**

* Can the authors comment on the practical computational overhead of the DP-GMM fitting process during training and, more importantly, the signature matching process during inference, especially as the number of tasks $m$ grows large (e.g., hundreds or thousands)?

* How robust is the GMM fitting and retrieval performance if the underlying embedding distributions deviate significantly from Gaussian mixtures? Have the authors explored alternative, potentially more robust, signature representations?

* The adaptive LoRA update combines knowledge transfer ($S_\tau$) and orthogonal new directions ($\Delta\omega_{k+1}^\perp$). Could the authors provide ablation studies isolating the effects of just the orthogonality constraint versus just the knowledge transfer component (compared to the baseline without either)?

* Theorem 3.5 suggests error can be made arbitrarily low if $\delta$ is large enough. How large does $\delta$ typically become in practice with the proposed adaptive fine-tuning? Does it continue to grow sufficiently as $m$ increases, or does it saturate, potentially limiting performance on very long sequences? (Figure 2 provides some insight for CIFAR-100, but more analysis would be helpful).

---

> ### Author Response · Authors · 2025-11-21
> **Official Comment**
>
> We would like to thank the reviewer for recognizing the novelty and significance of our work, particularly Theorems 3.4 and 3.5. Your other questions are addressed below.
>
> **High System Complexity:**
>
> **Reproducibility.** We have taken deliberate steps to ensure reproducibility and practical usability of our framework:
>
> We provide the complete source code, including all training, inference, and evaluation scripts.
> The submission includes detailed instructions for environment setup and execution.
> We provide a simple, streamlined interface for reproducing all of our experiments using a single command:  python main.py --cfg {config file} , with all configuration files available in the configs/ folder
>
> To further support reproducibility, we have documented all hyperparameters and implementation details in Appendix E (Datasets, Hyperparameters, and Pre-Trained Models section).
>
> Our method is fairly simple to implement as summarized below.
>
> **Implementation Complexity.** PROTEUS learns task-specific LoRA adaptation composed of frozen past directions and new orthogonal directions via solving Eq. (11). For each task-specific LoRA unit, it computes the corresponding input embeddings which are then partitioned into representation clusters using DP-GMM as summarized in Eq. (4). The approximated Gaussian mean and variance of each embedding cluster can be used as a signature of the learned LoRA.
>
> During test time, PROTEUS retrieves the cluster whose (approximated) Gaussian distribution assigns highest likelihood for each test input -- see Eq. (3) -- and uses its corresponding LoRA to represent the input for parameter-free classification -- see Eq. (12). The statistics used in Eq. (12) were computed during training in lines 10-11 of Algorithm 1.
>
> **Pseudocode.** The whole training and test procedures are summarized in Algorithm 1 (14 lines of pseudo-code); and Algorithm 2 (6 lines of pseudo-code in Appendix B).
>
> **Fragility Test.**  We provide extensive evaluation of our work on the Visual Task Adaptation Benchmark (VTAB) curated by Google Research. VTAB  comprises a challenging set of numerous downstream vision tasks, coming from diverse domains: natural images, artificial environments (structured) and images captured with non-standard cameras (specialized). Achieving good results (outperforming all SOTA methods) on this challenging benchmark suggests that our method has strong practical applicability (see Section 5 - Table 1, Figure 3.).
>
>
>
> **More Ablation of Design Choices**:
>
> **Advantage of DP-GMM over K-Means.** Conceptually, DP-GMM offers two key advantages over K-means clustering. First, it automatically adapts the number of clusters, which is crucial in continual learning where tasks vary widely in complexity and semantics. Second, DP-GMM’s probabilistic modeling allows PROTEUS to compute principled likelihood scores for test inputs, and leverage those for retrieval. To empirically demonstrate the importance of the DP-GMM module, we provide  an additional experiment  in which DP-GMM is replaced  with K-Means (using K=20) in our pipeline. This simple change  resulted in up to **13/18% drop** in classification/retrieval accuracy (see the table below).
>
> | Datasets          | DP-GMM (Classifier) | DP-GMM (Retrieval) | K-Means (Classifier) | K-Means (Retrieval) |
> |-------------------|----------------------|---------------------|-----------------------|----------------------|
> | Split ImageNet-R  | **82.69**            | **96.89**           | 81.15                | 92.75               |
> | Split ImageNet-A  | **63.77**            | **77.55**           | 55.34                | 63.63               |
>
> **Advantage of Parameter-Free Retrieval.** We have previously  ablated the contribution of our parameter-free retrieval to the overall performance. This is achieved via replacing it with **a simple retrieval scheme** that uses only the last task's parameters. Table 2 in our manuscript shows that, without our retrieval module, this method results in a **32% drop** in average performance compared to PROTEUS.

---

> ### Author Response · Authors · 2025-11-21
> **Official Comment**
>
> **Advantage of Enforcing Orthogonality and Sparsity of Selection Matrix in Eq. (11).** We have also provided ablation studies that validate the **importance of orthogonality of LoRA units and the sparsity of the selection matrix** (see Appendix F of our manuscript)
>
> - **Sparsity Regularizer.** To analyze the impact of this regularizer, we compared PROTEUS against its variant with the sparsity regularizer turned off via setting λ = 0 in Eq. (11). The performance of this variant is shown to degrade severely (see second last row of Table 6). This degradation can be attributed to the excessive and unfiltered reuse of irrelevant LoRA units, including those that negatively affect performance, making the model harder to optimize and reducing its effectiveness.
>
> - **Orthogonal Regularizer.** To analyze the impact of this regularizer, we had compared PROTEUS against another variant without orthogonal regularizer. This result is summarized in Figure 4 which shows that without this regularizer, the performance of PROTEUS consistently degrades across multiple benchmark datasets.
>
> **Reliance on Distributional Assumptions:**
>
> We acknowledge that real embedding distributions might not be Gaussian. It is however well-established that any distribution can be approximated well with a mixture of Gaussian with a sufficiently large number of Gaussians. This is enabled via the use of DP-GMM which is a non-parametric clustering method that automatically adapts the number of Gaussians to fit each distribution. This design allows the model to handle complex, non-Gaussian distribution approximately well as verified by the overall superior performance over existing SOTA baselines.
>
> Furthermore, our results on the most challenging VTAB benchmark that comprises a wide range of tasks with large semantic gaps (whose input embeddings are likely complex, non-Gaussian) show that PROTEUS consistently performs better than existing SOTA baselines (see Table 1). This attests to the reliability of our parameter-free (or non-parametric) retrieval mechanism even when the real embedding distributions are not Gaussian.
>
> Overall, while alternative signature representations could be explored, our DP-GMM approach provides a flexible and effective default, and extending the framework to fully non-Gaussian embeddings is a promising direction for future work.
>
> **Practical Computational Overhead**
>
> The complexity of PROTEUS inference is given in Appendix C.2 as $O(Tmd^3 + Cd^3)$, where m denotes the number of tasks, T the maximum number of Gaussian components, and C the number of classes (constant). The computational overhead thus increases linearly in terms of m and T. Fixing T=100, we additionally report the per-batch overhead (in seconds) on ImageNet-R as a function of the no. of tasks m:
>
> |Task 1|Task 2|Task 3|Task 4|Task 5|Task 6|Task 7|Task 8|Task 9|Task 10|
> |------|------|-------|-------|-------|-------|-------|-------|-------|-------|
> |0.41|0.77|1.03|1.31|1.66|2.13|2.42|2.60|2.97|3.29|
>
> In the context of long sequences of tasks, we run on CIFAR-100 with m = 50 tasks, 2 classes each task. The overhead (in seconds) each batch (batch size = 32) is a function of the no. of tasks m as below:
>
> |Task 2|Task 5|Task 10|Task 15|Task 20|Task 25|Task 30|Task 35|Task 40|Task 45|Task 50|
> |------|------|-------|-------|-------|-------|-------|-------|-------|-------|-------|
> |0.27|0.64|1.33|2.08|2.93|3.86|4.89|5.91|7.18|8.26|9.59|
>
>
> Overall, while there are indeed computational overheads, we believe these are not significant and can be afforded in exchange for much better performance.
>
> **How large does delta typically become in practice**
>
> As requested, we provide below additional validation on how large delta can become in practice, especially when the task sequence is long. We compare the empirical and theoretical delta along a sequence of 50 continual tasks derived from the CIFAR-100 dataset. Note that the theoretical delta serves as a lower-bound on cluster separation to guarantee that the maximum retrieval error is below 2%. As shown below, the empirical delta significantly exceeds the minimum requirement which confirms the validity and tightness of our theoretical result in practice.
>
> |-|Task 5|Task 10|Task 15|Task 20|Task 25|Task 30|Task 35|Task 40|Task 45|Task 50|
> |------|------|------|-------|-------|-------|-------|-------|-------|-------|-------|
> |Empirical delta|163.01|105.02|163.31|159.36|140.94|98.93|128.74|88.84|145.01|143.41|
> |Theoretical delta|5.17|6.60|5.67|5.88|5.71|6.24|4.68|6.31|5.40|6.59|

---

### Official Review · Reviewer_yqcf · 2025-10-31

**Soundness:** 3
**Presentation:** 2
**Contribution:** 3
**Rating:** 6
**Confidence:** 3

**Summary:**

The paper introduces PROTEUS, a method for parameter-based continual fine-tuning (CFT) that improves parameter-free retrieval. The method has two synergistic parts. First, an "Adaptive Knowledge Composition" strategy trains new LoRA adapters to be a selective combination of past adapters plus a new, orthogonal component for task-specific knowledge. Second, this orthogonal training creates highly separated representation clusters, which are captured by a "Parameter-Free Retrieval" mechanism (a DP-GMM). At test time, an input is assigned to the task with the highest likelihood—a non-learnable lookup. The paper provides theoretical bounds linking this cluster separation to low retrieval error and shows SOTA empirical results, demonstrating superior retrieval and predictive performance.

**Strengths:**

1. The paper provides a novel application of Gaussian Mlixutre Models on CF. Improving prior state of the art in CFT with LoRA-based adapters.
2. The paper provides a solid theoretical backing for its approach.
3. The paper provides empirical evidence of the consistently higher performance of their method compared to previous state of the art across a diverse set of datasets.

**Weaknesses:**

1. The paper claims to alleviate key issues with lack of representation adaptability and forgetting in retrieval-based methods. However, the latter is a prominent problem in prompt-based methods, not in parametric CFT methods (as PROTEUS) and previous parametric CFT approaches (RanPAC, InfLoRA, SD-LoRA) already address this with parameter-free retrieval methods.

**Questions:**

1. "Provably Accurate" is a term consistently mentioned in the paper but not properly introduced. Adding reference when mentioned for the first time would strengthen the readability of the paper.
2. I understand that at test time the input embedding for unseen samples has to go through each of the LoRA-augmented architectures in contrast to previous methods that used the first or last adaptation. How does the inference latency scale as we increase the number of tasks compared to the other parametric CFT baselines ?

---

> ### Author Response · Authors · 2025-11-21
> **Official Comment**
>
> We would like to thank the reviewer for the positive rating. Your questions are addressed below.
>
> **Retrieval in parametric CFT methods:**
>
> We would like to explain that representation retrieval has not been addressed in existing LoRA-based parametric CFT methods. Instead, these methods assume that semantically related tasks can share a single representation (often the first or last task’s LoRA) at test time, thereby eliminating the retrieval module to avoid forgetting.
>
> In practice, continual tasks typically have large semantic shifts, and hence often violate the above assumption. For LoRA-based methods, however, adding a retrieval component is non-trivial. This is because LoRA parameters do not live in the input embedding space, and thus there is no natural similarity measure to associate a test input with stored LoRA units at inference time (lines 160-161). To demonstrate this, we conducted an additional experiment on ImageNet-R, combining RANPAC (which is best among state-of-the-art parameter-adaptation methods) with the parametric retrieval mechanism of HiDE [1] to retrieve the most relevant LoRA unit for each input during test time.  As a matter of fact, our experiment shows that a direct integration of a widely used parametric retrieval procedure in prompt-based CFT to LoRA-based CFT results in performance degradation due to poor retrieval accuracy.
>
> |Method|Performance|Retrieval|
> |----------|---------------|-----------|
> |RanPAC (w/ Retrieval)|71.79|75.18|
> |RanPAC (wo/ Retrieval)|78.08| N/A |
> |PROTEUS|**82.17**|**96.02**|
>
> PROTEUS addresses this gap with a parameter-free retrieval mechanism: each task’s input embeddings generate multi-key signatures that serve as keys to retrieve corresponding adaptation modules. This design mitigates retrieval forgetting and allows the model to use adapted embeddings at test time, leading to consistently better performance across tasks (Tables 1-3, Figure 3). It is also guaranteed to achieve high retrieval accuracy (see Theorem 3.5) which inspires our algorithm design in Section 4. The theoretical bound on the retrieval accuracy of the designed algorithm  is also empirically demonstrated to be tight (see Fig. 2).
>
> To summarize, PROTEUS combines the strengths of both paradigms by designing a LoRA organization that enables **adaptive representation use** while performing **parameter-free, signature-based retrieval** that introduces no trainable retrieval parameters, and thus is less prone to forgetting. The **key novelty** here is that **we are the first to deliver a parameter-free retrieval design with provable guarantee on high retrieval accuracy.**
>
> **Provably Accurate Clarification:**
>
> We will add a clarification in our revision to make it clear that being provably accurate means achieving high retrieval accuracy with arbitrarily high probability as shown in Theorem 3.5. This clarification will be added to the first occurrence of “provably accurate” to improve clarity and readability. We thank the reviewer for the suggestion.
>
> **Inference Latency Scale:**
> The complexity of PROTEUS inference is given in Appendix C.2 as $O(Tmd^3 + Cd^3)$, where m denotes the number of tasks, T the maximum number of input embedding clusters (i.e., number of Gaussian components in DP-GMM), and C the number of classes (constant). The computational overhead increases linearly in terms of m and T. Fixing T=100 (the largest T in all settings), we report below the per-batch overhead (in seconds) on ImageNet-R as a function of the number of tasks m:
>
> |Task 1|Task 2|Task 3|Task 4|Task 5|Task 6|Task 7|Task 8|Task 9|Task 10|
> |------|------|-------|-------|-------|-------|-------|-------|-------|-------|
> |0.41|0.77|1.03|1.31|1.66|2.13|2.42|2.60|2.97|3.29|
>
> In the context of long sequences of tasks, we further run PROTEUS on CIFAR-100 with m = 50 tasks, 2 classes each task. The overhead (in seconds) each batch (batch size = 32) is a function of the no. of tasks m as below:
>
> |Task 2|Task 5|Task 10|Task 15|Task 20|Task 25|Task 30|Task 35|Task 40|Task 45|Task 50|
> |------|------|-------|-------|-------|-------|-------|-------|-------|-------|-------|
> |0.27|0.64|1.33|2.08|2.93|3.86|4.89|5.91|7.18|8.26|9.59|
>
> Overall, we believe the computational overheads are negligible and can be afforded in exchange for much better performance (see our results in Section 5 - Table 1, Figure 3).
>
> —----------------------------------------------------------------------------------------
>
> [1] Wang et al., “Hierarchical Decomposition of Prompt-based Continual Learning,” NeurIPS 2023.

---

### Official Review · Reviewer_4Hxf · 2025-11-11

**Soundness:** 2
**Presentation:** 1
**Contribution:** 2
**Rating:** 2
**Confidence:** 2

**Summary:**

The authors examine the strengths and weaknesses of input-adaptation (prompt-based) and parameter-adaptation (LoRA-based) approaches for continual fine-tuning of pre-trained models. Input-adaptation enables flexible test-time behavior but suffers from retriever forgetting, while parameter-adaptation is stable but lacks adaptability. To bridge this gap, they propose PROTEUS, a parameter-free, provably accurate task-retrieval framework that learns orthogonal low-rank adapters and retrieves the most relevant one via statistical matching of task-specific signature distributions. Theoretical analysis provides error bounds linking retrieval accuracy to cluster separation, and experiments on benchmarks such as CIFAR-100, ImageNet-R/A, and VTAB-5T show that PROTEUS achieves state-of-the-art performance with improved adaptability and minimal forgetting.

**Strengths:**

- The authors try to mathematically justify the proposed approach
- The proposed method outperforms the baselines

**Weaknesses:**

- The paper is poorly written and difficult to follow, even after multiple readings. It would benefit from substantial restructuring, rewriting, and clearer explanations throughout.

	- The distinction between input-adaptation and parameter-adaptation is unclear. From the paper, it seems that input-adaptation corresponds to prompt-tuning and parameter-adaptation to LoRA-based fine-tuning, but these categories are only briefly mentioned in the abstract and never clearly defined or elaborated upon in the main text.
	- The authors state that input-adaptation relies on retrieving relevant prompts at test time, whereas parameter-adaptation uses a fixed embedding function, is retrieval-free, and avoids forgetting. However, it is not well explained why these approaches necessarily have these characteristics. It seems the intended distinction is that one requires identifying task-specific parameters at test time while the other does not. If that is the case, it would be clearer to use standard terminology such as task ID and task-specific parameters instead.
	- Moreover, the claim that only input-adaptation requires test-time task retrieval while parameter-adaptation does not is too rigid. Whether retrieval is required depends on how task-specific parameters are constructed, not inherently on whether the method is input- or parameter-based.
	- The paper’s terminology is often confusing and inconsistent, making it hard to understand the authors’ intent. For example, the phrase "Solution Vision" in the "Existing Literature" section is ambiguous. It appears to mean "proposed solution." Similarly, terms like "signature pattern," "nearest signature," and "signature distribution" are introduced without clear definitions. Such vague or unconventional terminology significantly reduces the readability and clarity of the paper.

**Questions:**

Refer to my comments in weaknesses

---

> ### Author Response · Authors · 2025-11-21
> **Official Comment**
>
> We appreciate the reviewer’s interest in clarifying our key conceptual arguments.
>
> **1. Distinction between input- and parameter-adaptation.** In this work, we study continual learning in the fine-tuning setting (i.e., continual fine-tuning), where a pre-trained backbone is kept frozen throughout and all task-specific knowledge must be incorporated through lightweight adaptation. Existing approaches fall into two broad paradigms:
>
> - **Input adaptation** refers to adaptation methods that append learnable parameters (i.e., prompts) to the tokenized input sequences (e.g., sequences of image patches). Examples include prompt-tuning methods [1].
>
> - **Parameter adaptation** refers to methods that directly modify the model’s parameters or add small trainable parameter blocks to the model’s pre-trained parameters. Examples include LoRA-based methods [2].
>
> **2. Explaining design choices of prior input- and parameter-adaptation approach.** A key component in continual fine-tuning is a retrieval module that decides which past task-specific representation (e.g., prompts or LoRA units) to reuse when the model sees new inputs at test time.
>
> Existing input-adaptation approaches (i.e., continual prompt-tuning) use parametric retrieval modules [3,4,5], which are prone to forgetting as the retrieval parameterization is also being updated continually as new tasks arrive.
>
> Existing parameter-adaptation methods (i.e., continual low-rank adaptation) in contrast assume that when tasks are similar in semantic, it is possible to use the same representation for all tasks (e.g., first- or last task’s LoRA) during test time and remove the retrieval module entirely to avoid the risk of forgetting [6,7].
>
> **3. On the claim that only input-adaptation requires test-time task retrieval while parameter-adaptation does not.**
>
> We would like to emphasize that **we never made this claim**. Rather, we explain that prior parameter-adaptation works (such as RANPAC) made this claim and in contrast, we show that **retrieval remains essential, especially when the continual tasks have large semantic shifts.**
>
> Below, we highlight the novelty of our work (3.1), summarize its algorithmic workflow (3.2 and 3.3), and provide additional results ablating the empirical advantage of parameter-free retrieval over existing parametric retrieval on parameter-adaptation methods (3.4).
>
> **3.1 Our novelty is two-fold.**
>
> Our method, **PROTEUS**, is the first parameter-adaptation work with test-time retrieval. It introduces a LoRA organization that enables **adaptive test-time representation** while using a **parameter-free, signature-based** retrieval that is less prone to forgetting.
>
> Second, we are the first to deliver a parameter-free retrieval design with **provable guarantee on high retrieval accuracy**. This bound inspires the design of our practical algorithm and is shown to be empirically tight -- see Figure 2 in our main text.
>
> **3.2 Workflow.**
>
> PROTEUS learns task-specific LoRA adaptation composed of frozen past directions and new orthogonal directions via solving Eq. (11). For each task-specific LoRA unit, it computes the corresponding input embeddings which are then partitioned into representation clusters using DP-GMM as summarized in Eq. (4). The approximated Gaussian mean and variance of each embedding cluster can be used as a signature of the learned LoRA.
>
> During test time, PROTEUS retrieves the cluster whose (approximated) Gaussian distribution assigns highest likelihood for each test input -- see Eq. (3) -- and uses its corresponding LoRA to represent the input for parameter-free classification -- see Eq. (12). The statistics used in Eq. (12) were computed during training in lines 10-11 of Algorithm 1.
>
> **3.3 Pseudocode.**
>
> The whole train and test procedures are summarized in Algorithm 1 (14 lines of pseudo-code); and Algorithm 2 (6 lines of pseudo-code in Appendix B). We refer the reviewer to our method’s intuition description in Section 4 for more details.

---

> > ### Author Response · Authors · 2025-11-21
> > **Official Comment**
> >
> > **3.4 Empirical Significance of Parameter-Free Retrieval over Parametric Retrieval.**
> >
> > We note that none of the existing parameter-adaptation methods for continual fine-tuning adopts retrieval, as explained earlier in 2. Adding existing parametric retrieval on the other hand will not work out of straightforward integration. To demonstrate this, we conducted an additional experiment on ImageNet-R, combining RANPAC (which is best among state-of-the-art parameter-adaptation methods) with the parametric retrieval mechanism of HiDE [1] to retrieve the most relevant LoRA unit for each input during test time.
> >
> > Our new results show that this vanilla integration of parametric retrieval expectedly degrades the performance of RANPAC due to forgetting (as evidenced by weak retrieval accuracy). On the other hand, PROTEUS with guaranteed parameter-free retrieval (see Theorem 3.5 and Fig. 2) achieves 96% of retrieval accuracy and consequently, significantly better performance (see table below).
> >
> > |Method|Performance|Retrieval|
> > |----------|---------------|-----------|
> > |RanPAC (w/ Retrieval)|71.79|75.18|
> > |RanPAC (wo/ Retrieval)|78.08| N/A |
> > |PROTEUS|**82.17**|**96.02**|
> >
> > **4. Standardizing Terminologies.**
> >
> > We are grateful to the reviewer for the suggestions on standardizing terminologies. We would like to clarify our terminologies below:
> >
> > - The term **solution vision** simply refers to an **overview of our proposed approach**, motivated by the limitations of previous input-adaptation and parameter-adaptation methods.
> >
> > - The **signature pattern** of a LoRA unit is defined in Definition 3.1 (lines 216-217) as the mean and variance of a Gaussian cluster of input embeddings generated using that unit. The corresponding Gaussian distribution is then the **signature distribution**.
> >
> > - The **nearest signature** (at test time) denotes the signature pattern whose corresponding **signature distribution** assigns highest likelihood for a given test input -- see Eq. (3).
> >
> > We have incorporated these clarifications in our revised manuscript.
> >
> > —----------------------------------------------------------------------------------------
> >
> > [1] Lester et al., “The Power of Scale for Parameter-Efficient Prompt Tuning,” EMNLP 2021.
> >
> > [2] Hu, Edward J., et al. “Lora: Low-rank adaptation of large language models.” ICLR 1.2 (2022): 3.
> >
> > [3] Wang et al., “Learning to Prompt for Continual Learning,” CVPR 2022.
> >
> > [4] Wang et al., “Dualprompt: Complementary prompting for rehearsal-free continual learning,” ECCV 2022.
> >
> > [5] Smith et al., “CODA-Prompt: Continual Decomposed Attention-Based Prompting,” CVPR 2023.
> >
> > [6] McDonnell et al., “RanPAC: Random Projections and Pretrained Models for Continual Learning,” NeurIPS 2023.
> >
> > [7] Liang et al., “Inflora: Interference-free low-rank adaptation for continual learning,” CVPR 2024.

---

### Author Response · Authors · 2025-11-21
**Rebuttal Summary**

We thank the reviewers for their thoughtful comments and for recognizing the strengths of our work:

- **Clarity of presentation:** reviewer **as1a**: “clearly identifies the problem”, reviewer **JdHy**: “clear, interpretable, and scalable solution”.

- **Principled theoretical analysis:** reviewer **as1a**: “non-trivial theoretical analysis linking the retrieval error rate to geometric properties”,  reviewer **yqcf**: “provides a solid theoretical backing for its approach”, reviewer **JdHy**: “formally connects retrieval accuracy to structural properties of task-specific representation clusters”.

- **Strong empirical performance**: reviewer **4Hxf**: “the proposed method outperforms the baselines”,  reviewer **as1a**: “Achieves SOTA performance on several challenging CL benchmarks”.

Across the four reviews, the main requests for clarification center on:

- **Elaborating key terminologies**: input- and parameter-adaptation continual fine-tuning (CFT), signature pattern and distribution, and how these methods approach representation retrieval during test time (reviewer **4Hxf** and **yqcf**).

- **Explaining design choices**: system complexity, additional ablation studies, and assumptions in the theoretical analysis (reviewer **as1a** and **JdHy**).

- **Evaluating scalability**: stress test our approach in cases with longer task sequences (reviewer **yqcf** and **JdHy**).

We have addressed these points as follows:

**Elaborating key terminologies.**

- We explained **input adaptation, parameter adaptation**, as well as their distinction that leads to the conception of our framework. See our response to reviewer **4Hxf**.

- We clarified the terminologies **signature pattern, signature distribution**, and **nearest signature** in our response to reviewer **4Hxf**.

**Explaining design choices.**

- We performed an additional experiment to demonstrate the non-triviality of integrating a retrieval mechanism into a LoRA-based CFT workflow in our response to reviewer **4Hxf**.

- We discussed and ran additional ablation studies on our key design choices, including DP-GMM clustering, orthogonal LoRA updates, and parameter-free retrieval in our response to reviewer **as1a** and reviewer **JdHy**.

**Evaluating scalability.**

We conducted additional experiments to demonstrate the scalability of our method (i.e., inference latency, computational overhead) in scenarios with significantly longer task sequences (in response to Reviewers **yqcf, as1a**). We further demonstrate the robust performance of our work in memory-constrained scenarios with long task sequences (in response to Reviewer **JdHy**).

- Overall, our empirical results consistently show that PROTEUS remains robust and scalable with affordable cost under complex, real-world feature distributions, supporting the validity and practical strength of our design.

**Summary of our approach and contributions:**
Our method, **PROTEUS**, is a continual fine-tuning approach that combines the strengths of two existing CFT paradigms: input adaptation and parameter adaptation. It overcomes the limitations of both paradigms by introducing the **first parameter-adaptation framework with parameter-free retrieval**, enabling adaptive test-time behavior without being prone to forgetting.

This is achieved with learning task-specific LoRA adapters via solving Eq. (11), computing their induced input embeddings, and partitioning these embeddings into clusters. The representations of these clusters form the basis for our parameter-free retrieval mechanism and its theoretical guarantee on high retrieval accuracy during test time. This delivers a significant theoretical contribution as we are the first to provide provable retrieval-accuracy guarantee for parameter-adaptation in continual fine-tuning with explicit retrieval accuracy bound (see Theorems 3.4 and 3.5). Our rigorous analysis thus offers principled guidance to design a novel parameter-free retrieval mechanism that remains robust and highly and consistently effective in numerous experiment scenarios with large semantic drift across tasks, long task sequences, and even memory-constrained scenarios.

We thank the reviewers once again for their careful evaluations and look forward to engaging in discussion. We have also uploaded a revised manuscript.

---

### Author Response · Authors · 2025-12-03
**Closing Statement**

**Dear Area Chair**,

Thank you for overseeing the review of our submission. We understand this is an unusual situation that adds extra burden to the new AC and we sincerely appreciate your time.

We also thank reviewers **yqcf, as1a, and JdHy** for their original acceptance score (**6, 6, 6**), and reviewer **JdHy** particularly for highlighting our presentation clarity (“clear, interpretable, and scalable solution”). We appreciate that reviewer **4Hxf** initially found the paper challenging  to read but still provided a precise and accurate summary that highlighted our novelty well.

To ease the extra burden for the new AC, we would like to provide a more concise summary of our key contributions and reviewer suggestions which we have addressed in our rebuttal.

**1. Key Contributions and Novelty**

**Theoretical Contribution**. Our work introduces a continual fine-tuning (CFT) framework (**PROTEUS**) which features the **first** probably approximately correct **(PAC) parameter-free retrieval mechanism**. It guarantees that during test time, the **correct representation adaptation** module (i.e., LoRA) **is retrieved for each input with high probability** (Theorems 3.4 and 3.5). It is also shown that the PAC bound is **numerically tight** (Fig. 2).

**Idea Novelty.** The above distinguishes our work from prior CFT approaches which either (1) adopt a parametrized representation retrieval and risk forgetting as the retrieval mechanism is continually updated; or (2) assume strong task similarity and use a fixed representation mechanism across tasks, thus ignoring the forgetting risk. The theoretical insight also inspires a new design for CFT which develops a **novel signature-based** non-parametric **clustering** of task-specific representation adaptation modules that induces a **large cluster separation** that **tightens the PAC retrieval mechanism** to provide **at least 96% of retrieval accuracy**.

**Algorithmic Contribution.** This is achieved via (i) learning each task-specific LoRA adaptation composed of frozen past directions and new orthogonal directions via solving Eq. (11) and (ii) computing its corresponding input embeddings which are then partitioned into representation clusters using DP-GMM as summarized in Eq. (4). The approximated Gaussian mean and variance of each embedding cluster can be used as a signature of the learned LoRA (see Definition 3.1). During inference, an input queries the nearest signature with the highest likelihood (lines 208–209), enabling accurate task retrieval without adding any retriever parameters and risking forgetting.

**2. Practical Impact**

Unlike prior CFT which either ignores representation retrieval or uses forgetful retrieval mechanisms, our proposed CFT (named **PROTEUS**) enables retrieval with provably low forgetting, thus **achieving significantly better performance than all existing CFTs.**

Our experiment has demonstrated this across a wide range of benchmark datasets (CIFAR100, ImageNet-A, ImageNet-R, VTAB5T-Large, VTAB5T-Small, VTAB-SIM50) with various levels of semantic drift (mild, abrupt, varying) across tasks and lengths of task sequences (5-100).

Across all scenarios, PROTEUS consistently surpasses competing baselines, with especially large improvements when cross-task semantics differ substantially (Tables 1 & 3, Fig. 3). As suggested by reviewers **yqcf and as1a**, we have also conducted  ablation experiments to show that PROTEUS indeed maintains strong performance in scenarios with long task sequences (50–100 tasks). We also added **computational and memory analyses** as requested by the reviewers which show that our method achieves much better performance at a small extra processing overhead.

Overall, **PROTEUS is the first PAC parameter-free retrieval mechanism for continual fine-tuning (CFT)** which establishes a new framework for provable CFT design that can benefit future research in this field.

**3. Review Summary and Rebuttal Resolution**

In the detailed review, the reviewers request the followings:

- **Elaborating some terminologies:** input- and parameter-adaptation continual fine-tuning (CFT), signature pattern and distribution, and how these methods approach representation retrieval during test time (reviewer **4Hxf and yqcf**).

- **Explaining design choices:** system complexity, additional ablation studies, and assumptions in the theoretical analysis (reviewer **as1a and JdHy**).

- **Evaluating scalability:** stress test our approach in cases with longer task sequences (reviewer **yqcf and JdHy**).

We have addressed all the above with extensive additional experiments and ablation studies which have all been incorporated into our updated manuscript. These changes have also been summarized in the previous message.

**4. Closing**

Thank you again for your time and consideration. We understand this phase is not a discussion, but we are available for any follow-up questions from the AC.

Sincerely,

Authors.

---

### Meta-Review · Area_Chair_ePGh · 2026-01-19

**Summary:**

Disclaimer: I read the revised paper carefully by myself prior to checking the reviewers' comments and the authors' rebuttal.

The decision was mainly driven by unresolved issues of clarity and rigor. The paper still lacks a clear problem formulation and a precise, well-founded distinction between input-adaptation and parameter-adaptation; the rebuttal does not move beyond a largely hand-waving discussion. This also affects the positioning of the work, as the pros and cons relative to existing families of methods are not clearly articulated.

Additionally, the claim of “provably accurate” remains unconvincing: the stated theorem appears disconnected from the actual algorithm and reads as an exaggerated guarantee. In particular, Theorem 3.5, which is a elementary probability two-Gaussian overlapping problem, at best hints that delta, the separation factor, plays a role in the error analysis. But the whole Section 4 (algorithm design) does not discuss why delta is connected to the algorithm design in equation (11). The only evidence throughout Section 4 is the emprical evaluation, which trivially shows that the empirical delta is much larger than the theoretical lower bound (well, this can hardly be called "tight"). This is far from a "provable" algorithm as promised in the title and abstract.

Concerns about strong distributional assumptions (e.g., reliance on GMMs) are technically unresolved, with no discussion of robustness beyond Gaussian mixtures, though this could be deferred to future work.

In contrast, issues regarding inference-time complexity and ablation of design choices were mostly addressed, but not enough to offset the core concerns above.

**Reviewer Concerns:**

* poor writing: not addressed by the rebuttal. The distinction between input-adaptation and parameter-adaptation is still only hand-wavingly discussed, without a sound definition. In fact, the problem setting is not even properly defined (e.g. what's the goal of the problem, mathematically?) I tend to agree with the reviewer that the characteristics between input-adaptation and parameter-adaptation, and how the proposed approach obtain the best characteristic from each side, is far from clear.

* improper discuss of the cons/pros to existing families of works: While the authors discuss in the rebuttal about the difficulty of Retrieval in parametric CFT, the bigger picture of the cons/pros of existing families (echoing the previous discussion on clarity) has room for improvement.

* the definition of "provably accurate": With the authors' explanation, the theorem appears to be an exaggerated one isolated from the algorithm. So this is not solved.

* reliance on distributional assumptions: Technically this is not solved (no non-mix-Gaussian alternatives studied nor discussed). In particular, this issue is not addressed "How robust is the GMM fitting and retrieval performance if the underlying embedding distributions deviate significantly from Gaussian mixtures?" But perhaps it's fine to make it a future work.

* inference time and/or high system complexity: This is more or less resolved.

* ablation of design choices: This is more or less resolved.

**Reviewer Scores:**

* 4Hxf would at best change from 2 to 3 or stay as 2.
* yqcf would probably stay as 6, if not less.
* as1a may change from 6 to 7 or stay as 6.
* JdHy may change from 6 to 7 or stay as 6.

---

### Decision · Program_Chairs · 2026-01-26

Reject